# Establishing the role of ATP for the function of the RIG-I innate immune sensor

David C Rawling[1†], Megan E Fitzgerald[2†], Anna Marie Pyle[3,4]*

[1]Department of Molecular Biophysics and Biochemistry, Yale University, New Haven, United States; [2]Department of Molecular, Cellular and Developmental Biology, Howard Hughes Medical Institute, Yale University, New Haven, United States; [3]Department of Molecular, Cellular and Developmental Biology, Yale University, New Haven, United States; [4]Department of Chemistry, Howard Hughes Medical Institute, Yale University, New Haven, United States

**Abstract** Retinoic acid-inducible gene I (RIG-I) initiates a rapid innate immune response upon detection and binding to viral ribonucleic acid (RNA). This signal activation occurs only when pathogenic RNA is identified, despite the ability of RIG-I to bind endogenous RNA while surveying the cytoplasm. Here we show that ATP binding and hydrolysis by RIG-I play a key role in the identification of viral targets and the activation of signaling. Using biochemical and cell-based assays together with mutagenesis, we show that ATP binding, and not hydrolysis, is required for RIG-I signaling on viral RNA. However, we show that ATP hydrolysis does provide an important function by recycling RIG-I and promoting its dissociation from non-pathogenic RNA. This activity provides a valuable proof-reading mechanism that enhances specificity and prevents an antiviral response upon encounter with host RNA molecules.

*For correspondence: anna.pyle@yale.edu

†These authors contributed equally to this work

Competing interests: The authors declare that no competing interests exist.

## Introduction

Retinoic acid-inducible gene I (RIG-I) is a cellular innate immune receptor that recognizes viral ribonucleic acid (RNA) in the cytoplasm and consequently initiates a host defense response (*Yoneyama et al., 2004*; *Ablasser et al., 2009*; *Baum et al., 2010*). The RIG-I protein is comprised of three major domain groups, including tandem caspase activation and recruitment domains (CARDs) that mediate signaling, a modified DExD/H-box ATPase core, and a C-terminal domain (CTD) that provides RNA ligand specificity through high-affinity interaction with blunt-ended, triphosphorylated duplexes (*Hornung et al., 2006*; *Lu et al., 2010*; *Wang et al., 2010*; *Jiang et al., 2011*; *Kowalinski et al., 2011*; *Luo et al., 2011*, *2012b*; *Rawling et al., 2015*). RIG-I surveys the cytoplasm in an autorepressed conformation, with the CARDs stacked against the helicase core (*Kowalinski et al., 2011*; *Zheng et al., 2015*) (*Figure 1*). Upon RNA ligand recognition, the protein adopts a conformation that can bind and hydrolyze ATP (*Yoneyama et al., 2004*; *Cui et al., 2008*; *Gee et al., 2008*; *Jiang et al., 2011*; *Kowalinski et al., 2011*; *Luo et al., 2011*). These combined activities result in liberation of the CARDs for interaction with the adapter protein MAVS, causing MAVS oligomerization and subsequent downstream signaling (*Hou et al., 2011*; *Peisley et al., 2014*; *Rawling and Pyle, 2014*) (*Figure 1*).

RIG-I requires both RNA and ATP ligands in order to become activated and initiate signaling. The precise molecular determinants for functional RNA recognition by RIG-I have been well characterized through structural, biochemical and cell-based approaches (*Schlee et al., 2009*; *Luo et al., 2012b*; *Kohlway et al., 2013*; *Goubau et al., 2014*). However, the specific roles of ATP binding and hydrolysis in activating RIG-I for immune signaling have not been defined or differentiated. Mutational studies have underscored the key role of conserved amino acids within the ATPase active-site of RIG-I.

**eLife digest** When a virus invades a cell, it commandeers the cell's replication machinery to make copies of the virus' genetic material. Some viruses, such as those that cause influenza or measles, store their genetic information in the form of ribonucleic acid (RNA) molecules.

When a virus is first detected inside an animal, specialized cells are activated and sent to destroy the invader. This is known as the innate immune response. Animal cells contain a sensor known as RIG-I, and when RIG-I detects and binds to viral RNA, it starts a signaling process that activates the innate immune system. RIG-I can also bind to RNA molecules made by the host cell, but this binding does not cause RIG-I to activate the immune response.

Although there have been many studies into how RIG-I tells the difference between cell and virus RNA (this process is also known as 'proofreading'), many of these have overlooked the role of a molecule called ATP. ATP stores energy for the cell, which is released in a process called hydrolysis. Binding to RNA causes the shape of RIG-I to change so that it can also bind to ATP, and RIG-I cannot signal to trigger an immune response unless it is bound to ATP.

Now, Rawling, Fitzgerald and Pyle have investigated ATP's role in immune signaling in more detail, and have found that ATP plays two distinct roles. Binding to ATP is necessary for RIG-I to start signaling in response to viral RNA, as it activates or 'springs open' the signaling regions formed when RIG-I binds to RNA. ATP hydrolysis is not involved in signaling; instead, it helps to remove RIG-I from the cell's RNA molecules. This recycles RIG-I and prevents it from becoming activated at the wrong time.

With a clear mechanistic description of RIG-I proofreading in place, it will now be possible to investigate how certain viruses take advantage of this system to evade detection. Further investigations could also look at how the dysregulation of RIG-I proofreading may be related to autoimmune disorders and cancer.

For example, residues that are required for ATP hydrolysis, such as the conserved lysine within helicase motif I (the Walker A motif), or the conserved glutamate and aspartate residues within motif II, exhibit dramatically reduced signaling capabilities (*Yoneyama et al., 2004*; *Bamming and Horvath, 2009*; *Rawling et al., 2015*). In contrast, RIG-I signaling is not impaired by motif III mutations that significantly perturb ATPase activity (*Bamming and Horvath, 2009*), thereby making it difficult to establish a direct relationship between hydrolysis and signaling. Alternative roles proposed for ATP hydrolysis include translocation and filament formation (*Myong et al., 2009*; *Patel et al., 2013*; *Peisley et al., 2013*), however these functions have not been shown to be absolutely necessary for signaling by RIG-I (*Kohlway et al., 2013*; *Louber et al., 2014*). Thus, the function of ATPase activity in RIG-I signaling remains unclear.

To fully understand the determinants for RIG-I signal activation, the roles of ATP binding and hydrolysis must be established. While robust signaling is often associated with ATP hydrolysis activity (*Yoneyama et al., 2004*; *Bamming and Horvath, 2009*; *Luo et al., 2011*; *Kohlway et al., 2013*; *Patel et al., 2013*; *Peisley et al., 2013*; *Rawling et al., 2015*), the ability of an RNA ligand to stimulate RIG-I ATPase activity does not necessarily ensure immune-stimulatory activity. It has been demonstrated that duplex RNAs containing a 5′ hydroxyl can bind RIG-I and stimulate robust ATP hydrolysis (*Kohlway et al., 2013*; *Rawling et al., 2015*), yet these RNAs do not induce immune signaling by RIG-I, even at saturating concentrations (*Schlee et al., 2009*; *Baum et al., 2010*; *Goubau et al., 2014*). Similarly, RIG-I ATPase activity can be stimulated by short RNAs bearing a 3′ overhang, as seen in miRNA precursors (*Schlee et al., 2009*; *Wilson and Doudna, 2013*). There are many other cellular RNAs that might be expected to bind RIG-I, such as lncRNAs, which contain abundant stem-loop structures (*Somarowthu et al., 2015*) and 5′ monophosphorylated products of mRNA decapping (*Coller and Parker, 2004*). Therefore, a mechanism to discriminate between endogenous and pathogenic RNA must be utilized. The ability to differentiate between host and viral RNA is essential, as aberrant activation of RIG-I would result in unregulated signaling that could lead to auto-inflammation and other innate immune disorders. Thus, elucidating the mechanism for RNA ligand discrimination and understanding its linkage with ATP utilization are central to understanding the induction and regulation of RIG-I signaling.

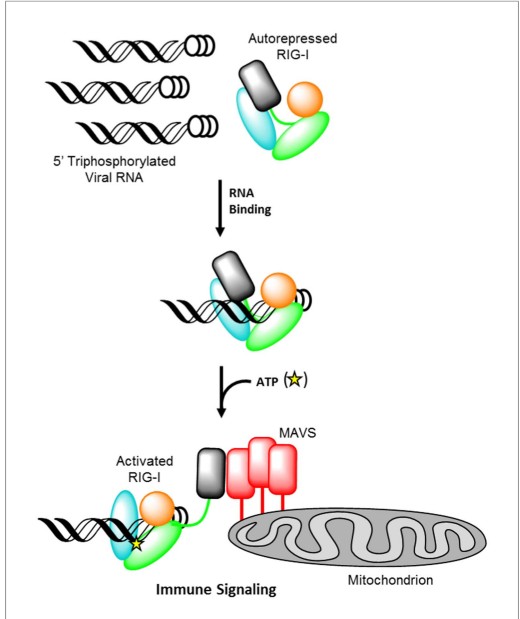

**Figure 1**. Retinoic acid-inducible gene I (RIG-I) signaling activation by viral ribonucleic acid (RNA). The RIG-I protein is comprised of four major domain groups including the caspase activation and recruitment domains (CARDs) (black), and RNA-stimulated ATPase core (green), a helical regulatory and binding domain inserted into the second lobe of the ATPase called HEL2i (cyan), and a triphosphate recognition and RNA binding domain at the c-terminus annotated the C-terminal domain (CTD) (orange). RIG-I is normally present in cells in an autorepressed conformation with the CARDs stacked against the HEL2i domain. Upon infection by a subset of RNA and DNA viruses, RIG-I binds 5′ triphosphorylated duplex termini of viral RNA (black helix with three white circles) deposited or transcribed in the cytoplasm. RNA binding stimulates ATP (yellow star) binding and hydrolysis by RIG-I to ADP (yellow triangle). At some point in this process, RIG-I becomes competent to engage an immune signaling response through an interaction with the adaptor protein MAVS (red), however the determinant step for signaling has not been conclusively demonstrated. Thus, the transition of RIG-I from autorepressed to signaling competent may occur at the stage of RNA binding, ATP binding, or ATP hydrolysis, and this is denoted here with arrows bearing question marks leading from each stage in RNA-stimulated ATPase activity to the immune signaling interaction.

In this work, we use a series of biophysical and cell-based approaches, together with mutational analysis, to distinguish between the roles of ATP binding and ATP hydrolysis in RIG-I function. We show that RNA-stimulated ATP binding can activate RIG-I for immune signaling in the absence of hydrolysis, thus explaining the lack of a correlation between ATP hydrolysis and signaling. Using a novel RNA ligand lacking free ends, we establish that ATP-mediated signal activation requires RNA end capping by the CTD. We demonstrate that in the absence of end capping, ATP binding and hydrolysis promote RIG-I dissociation from RNA rather than signaling, thereby contributing to proofreading and rejection of sub-optimal RNA ligands, such as endogenous RNA. Additionally, we show that ATP-stimulated RNA dissociation is dependent on the CARD domains of RIG-I. We therefore propose a model in which ATP binding by RIG-I leads to conformational changes that can result in either the liberation of the CARDs for signaling or RIG-I dissociation from RNA, depending on the affinity of RIG-I for the bound RNA ligand. ATP binding and hydrolysis thus play important roles in RIG-I function by activating signaling and ensuring RNA target specificity.

## Results

### A dsRNA 10-mer stimulates signaling in ATPase-deficient RIG-I

To investigate the relationship between ATPase activity and immune signaling in RIG-I, we evaluated the catalytic and signaling capabilities of RIG-I ATPase site mutants. Lysine 270, a conserved residue in the Walker A motif that coordinates ATP (*Walker et al., 1982*; *Yoneyama et al., 2004*; *Cordin et al., 2006*), was mutated to either an alanine or arginine. Mutation of this lysine residue typically abolishes both ATP binding and hydrolysis in related DEAD-box proteins, making mutants at this conserved residue useful for studying the role of ATPase activity. Using a coupled ATPase assay (*Luo et al., 2011*; *Kohlway et al., 2013*), we determined the steady state ATP hydrolysis activity of RIG-I and the lysine mutants (K270A and K270R) when activated by a 5′ triphosphory-

lated, 10 bp hairpin (5′ppp10L). In agreement with previous studies on this and similar DExD/H-box proteins, we found that the RNA-stimulated ATPase activity of both mutations was eliminated, while wild type RIG-I activity was robust (*Figure 2A*, *Table 1*) (*Yoneyama et al., 2004*; *Cordin et al., 2006*). Next, we measured the RNA binding affinity of K270A and K270R mutants for a FAM-labeled 10-mer RNA hairpin using fluorescence polarization, and found that both mutants bind RNA with affinities similar to the wild type protein (*Table 1*). Thus, we conclude that the observed ATP hydrolysis deficiencies are not a result of impaired RNA binding by these mutants.

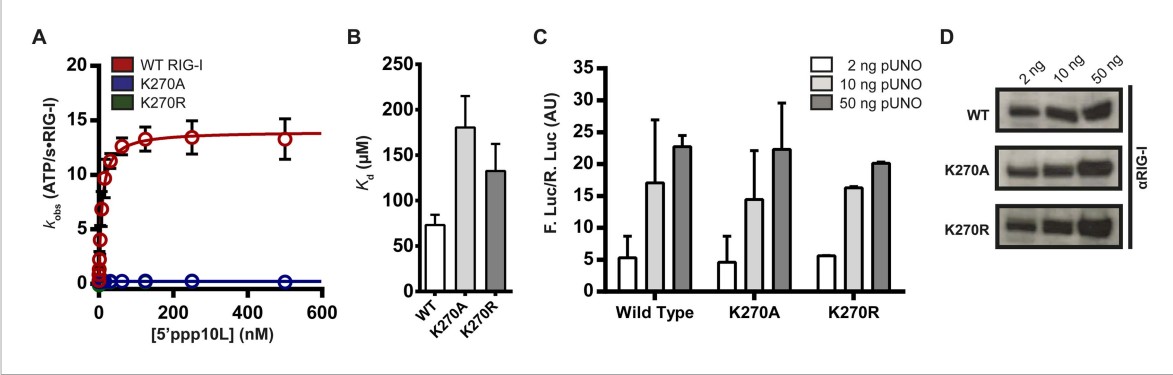

**Figure 2**. Walker A mutant RIG-I can induce IFN-β production in cells. (**A**) Steady state ATP hydrolysis by wild type, K270A and K270R RIG-I proteins stimulated with varying concentrations of the RNA hairpin 5′ppp10L. (**B**) Extrapolated dissociation constants for MANT-ATP binding by wild type and mutant RIG-I proteins bound to 5′ppp10L. (**C**) IFN-β induction in HEK 293T cells transfected with the indicated amount of the constitutive expression plasmid pUNO-hRIG-I containing either wild type or mutant RIG-I constructs. Cells expressing the indicated construct were challenged by transfection of 5′ppp10L. (**D**) Anti-RIG-I Western blot from HEK cell lysates. Plotted values are mean ± SD (n = 3).

The following figure supplement is available for figure 2:

**Figure supplement 1**. Determining association rate constants for MANT-ATP binding by Wild Type and Lysine 270 mutant RIG-I.

To determine if an inability to bind nucleotide by K270A and K270R causes the observed inhibition of hydrolysis, we measured ATP binding by WT and mutant RIG-I. To evaluate ATP binding, we monitored the association of WT and mutant RIG-I with a fluorescent ATP analog, MANT-ATP (*Figure 2—figure supplement 1G*), in the presence of the 5′ppp10L ligand using stopped flow fluorescence spectroscopy (*Figure 2—figure supplement 1A–F*). Using the observed rate constants, we extrapolated apparent dissociation constants ($K_d$) for MANT-ATP for each RIG-I construct (*Figure 2B*, *Table 1*, *Figure 2—figure supplement 1D–F*). Intriguingly, we found that Walker A mutants bind MANT-ATP with only a twofold weaker affinity than wild type RIG-I when stimulated by the 5′ppp10L RNA (*Figure 2B*). Thus, mutations in the ATPase active site specifically disrupt the chemical mechanism of hydrolysis, but they do not significantly perturb ATP binding by RIG-I when bound to 5′ppp10L.

We next evaluated the ability of K270A and K270R to stimulate IFN-β promoter activation using a cell-based assay system described previously (see 'Materials and methods', *Murali et al., 2008*; *Luo et al., 2011*). Briefly, HEK 293T cells are transfected with RIG-I or RIG-I mutant and an IFN-β/firefly luciferase reporter construct, challenged with RNA ligands, and then analyzed for luciferase expression (i.e., IFN-β promoter activation) using a luminescence assay. Several concentrations of each protein expression plasmid were transfected prior to dsRNA challenge in order to compare signaling efficiencies across wild type and mutant constructs. Intriguingly, wild type and Walker A mutant RIG-I proteins elicited similar levels of IFN-β promotor response at each of three concentrations tested when challenged by the 5′ppp10L hairpin (*Figure 2C*, *Table 1*). We performed Western blot analysis on cell lysates from IFN-β activation experiments and confirmed that wild type signaling levels for each mutant protein are not due to variable protein expression across the three constructs (*Figure 2D*). We also evaluated IFN-β promotor activation when cells were challenged by a vehicle control, and observed no response. We thus determined that the observed IFN-β activation requires RNA transfection, and is not the result of high cellular concentrations of RIG-I. Therefore, signaling by the Walker A mutants represents a *bona fide* response to exogenous RNA, and we conclude that ATP hydrolysis is not required for signaling by RIG-I on a high-affinity RNA ligand.

## ATP binding, not hydrolysis, is required for immune signaling

Previous studies have shown that RIG-I constructs containing mutations at lysine 270 will not induce a signaling response in cells (*Yoneyama et al., 2004*; *Bamming and Horvath, 2009*), in direct contrast to the results shown above. As the previous studies used longer dsRNAs to challenge RIG-I, we next

**Table 1.** Measured physical and biochemical constants

| Construct | WT RIG-I | K270A | K270R | ΔCARDs |
|---|---|---|---|---|
| Steady-state ATPase activity | | | | |
| 5′ppp10L | | | | |
| $k_{cat}$ (ATP/RIG-I•s) | 14.1 ± 0.4 | inactive | inactive | – |
| $K_{M, RNA}$ (nM) | 8.0 ± 1.0 | inactive | inactive | – |
| 5′ppp14L | | | | |
| $k_{cat}$ (ATP/RIG-I•s) | 14 ± 0.3 | inactive | inactive | – |
| $K_{M, RNA}$ (nM) | 15 ± 1.4 | inactive | inactive | – |
| 5′ppp30L | | | | |
| $k_{cat}$ (ATP/RIG-I•s) | 7.0 ± 0.2 | inactive | inactive | – |
| $K_{M, RNA}$ (nM) | 11 ± 1.4 | inactive | inactive | – |
| 5′ppp50L | | | | |
| $k_{cat}$ (ATP/RIG-I•s) | 6.5 ± 0.5 | inactive | inactive | – |
| $K_{M, RNA}$ (nM) | 50 ± 16 | inactive | inactive | – |
| Dumbbell | | | | |
| $k_{cat}$ (ATP/RIG-I•s) | 21 ± 1.0 | – | – | – |
| $K_{M, RNA}$ (μM) | 9.3 ± 1.4 | – | – | – |
| Maximum IFN-β production | | | | |
| 5′ppp10L (Firefly Luc/Renilla Luc) | 23 ± 1.8 | 22 ± 7.3 | 20 ± 0.2 | – |
| 5′ppp14L (Firefly Luc/Renilla Luc) | 16 ± 5 | 10 ± 0.1 | 21 ± 0.5 | – |
| 5′ppp30L (Firefly Luc/Renilla Luc) | 16 ± 7 | 0.5 ± 0.2 | 0.5 ± 0.2 | – |
| 5′ppp50L (Firefly Luc/Renilla Luc) | 46 ± 18 | 0.4 ± 0.1 | 0.6 ± 0.1 | – |
| 5′OH10L (Firefly Luc/Renilla Luc) | 3.7 ± 1.1 | 3.2 ± 0.1 | 8.9 ± 0.4 | – |
| Dumbbell (Firefly Luc/Renilla Luc) | 2.1 ± 0.5 | – | – | – |
| MANT-ATP binding | | | | |
| 5′ppp10L (μM) | 72 ± 4 | 183 ± 8 | 130 ± 30 | – |
| 5′ppp30L (μM) | 136 ± 6 | – | – | – |
| Equilibrium RNA binding | | | | |
| 5′OH10L (nM) | | | | |
| no nucleotide | 17 ± 1.0 | 42 ± 11 | 30 ± 4 | 19 ± 1.0 |
| with ATP | – | 83 ± 7 | 82 ± 2 | – |
| with ATPγS | 50 ± 5.0 | 98 ± 8.4 | 66 ± 6.1 | 28 ± 0.7 |
| with ADP-AlF$_4$ | 34 ± 4.9 | 60 ± 19 | 50 ± 9.0 | 21 ± 3.3 |
| with ADP | 32 ± 2.0 | 68 ± 16 | 53 ± 4.6 | 24 ± 2.3 |
| Dumbbell (μM) | 2.1 ± 0.2 | – | – | – |

Physical and biochemical constants for binding, catalysis, and cell-based signaling assays performed for wild type, Walker A-mutant, and ΔCARDs truncation constructs of RIG-I. Values represent mean ± SD (n = 3). See 'Materials and methods' for details of the experimental setups and data analysis used in each assay.

CARDs, caspase activation and recruitment domains; RNA, ribonucleic acid.

investigated whether the ATP binding and signaling activities of RIG-I are RNA length dependent. We performed the cell based analysis described above with wild-type RIG-I and the Walker A mutants using RNA ligands of increasing length, including 14-, 30-, and 50-mer duplex hairpins (5′ppp14L, 5′ppp30L, and 5′ppp50L respectively).

Wild type RIG-I exhibits robust signaling behavior when challenged by any of the RNA ligands tested. In contrast, the K270A mutant exhibits a significant loss in IFN-β activation (about twofold) when challenged by the 14-mer, and a complete loss in signaling when challenged by a 30- or 50-mer

RNA (*Figure 3A–C*, *Table 1*). For the K270R mutant, levels of IFN-β activation elicited by the 14-mer are comparable to those observed for the 10-mer; however, this mutant also exhibited no signaling when challenged by either the 30- or 50-mer hairpins (*Figure 3A–C*). Thus, Walker A mutations in RIG-I are detrimental to signaling only on longer RNA ligands.

To investigate the relationship between ATPase activity and immune signaling on these longer RNAs, we measured the ATP hydrolysis activity of RIG-I when it is stimulated by RNAs of varying duplex length, using the coupled ATPase assay (*Luo et al., 2011*; *Kohlway et al., 2013*). Wild type RIG-I exhibits RNA-stimulated ATPase activity on all RNA ligands, although the rates of catalysis ($k_{cat}$) vary considerably with duplex length (*Figure 3D–F*). For example, while 5'ppp14L stimulates ATP hydrolysis with a rate constant of ~15 s$^{-1}$, the analogous 30-mer displays a rate constant of only 8 s$^{-1}$. But, both RNA hairpins elicit nearly identical levels of IFN-β promoter induction, underscoring the lack of a direct apparent correlation between ATP hydrolysis and immune signaling. For the K270A and

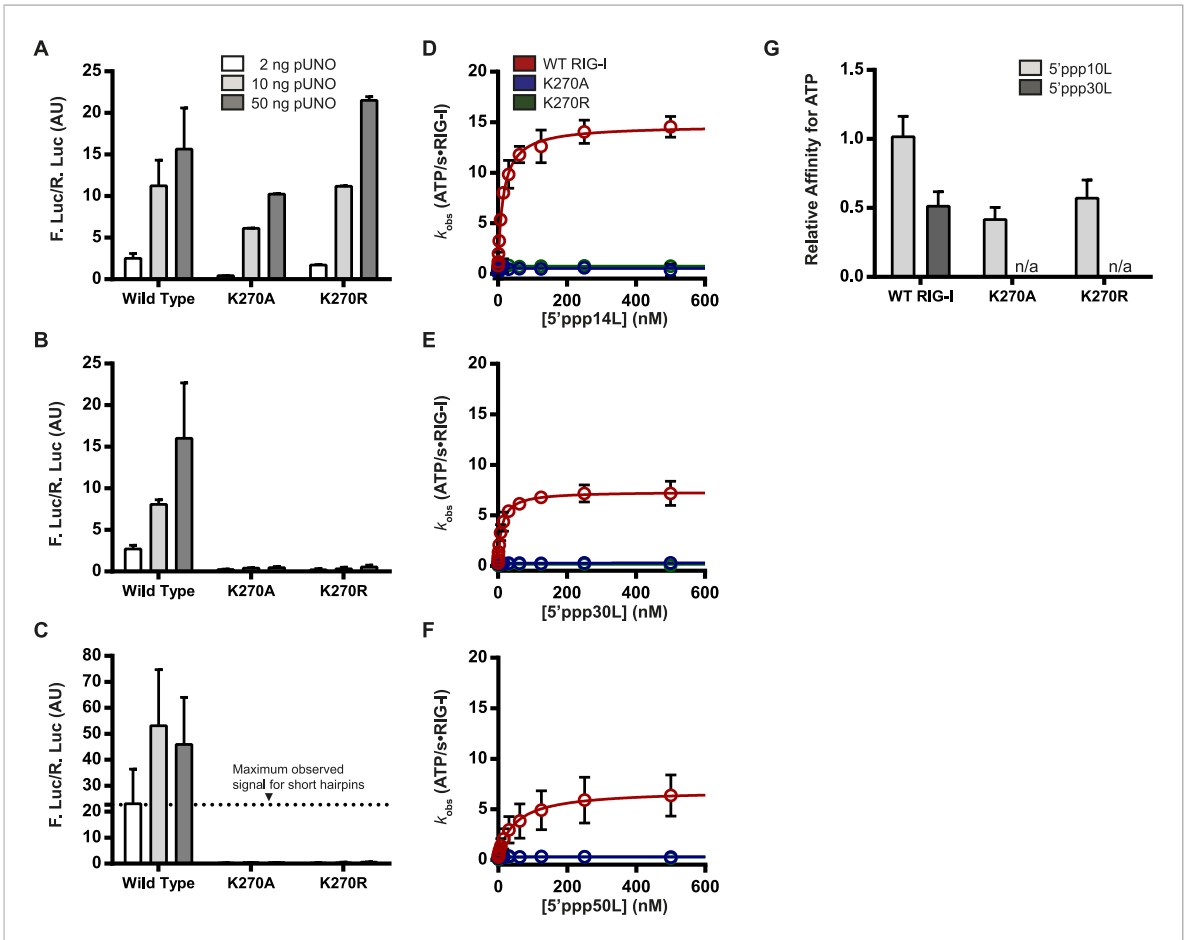

**Figure 3**. The requirement for ATP in IFN-β promoter induction is ligand dependent. (**A–C**) IFN-β induction in HEK 293T cells transfected with the indicated amount of the constitutive expression plasmid pUNO-hRIG-I containing either WT or mutant RIG-I constructs. Cells expressing the indicated construct were challenged by transfection of (**A**) 5'ppp14L, (**B**) 5'ppp30L, or (**C**) 5'ppp50L. (**D–F**) Steady state ATP hydrolysis by wild type and mutant RIG-I proteins stimulated with varying concentrations of the RNA hairpin (**D**) 5'ppp14L, (**E**) 5'ppp30L, or (**F**) 5'ppp50L. (**G**) Relative affinities for MANT-ATP binding by wild type and mutant RIG-I proteins bound in the context of 5'ppp10L, 5'ppp30L normalized to WT RIG-I bound to 5'ppp10L (*Figure 1B*). Plotted values are mean ± SD (n = 3).

The following figure supplements are available for figure 3:

**Figure supplement 1**. RIG-I forms multimers on 5'ppp50L at high protein concentrations.

**Figure supplement 2**. Assessment of RNA quality.

K270R mutants, all RNAs failed to stimulate measurable ATPase catalytic activity (*Figure 3D–F*, *Table 1*).

We subsequently measured the ATP binding affinities of WT RIG-I and mutant proteins when bound to 5′ppp30L. In contrast to the 5′ppp10L-bound mutants, 5′ppp30L did not stimulate ATP binding by either K270A or K270R (*Figure 3G*). By comparison, WT RIG-I is able to bind ATP in all cases, although its affinity for ATP weakens by two-fold with increasing RNA length (*Figure 3G*, *Figure 2—figure supplement 1A,D*). These experiments show that the only difference between Walker A mutants that are bound to 5′ppp10L and 5′ppp30L is their ability to bind ATP, and this ATP binding ability is directly correlated with immune signaling. We therefore conclude that RNA-stimulated ATP binding is required to induce signaling by RIG-I.

## RIG-I cannot signal when bound to internal duplex sites

Walker A mutants can bind ATP and induce immune signaling only when interacting with 5′ppp10L and not the longer 5′ppp30L, suggesting that RNAs containing off-target, internal duplexes are detrimental to RIG-I activity. To directly investigate the importance of RIG-I binding at internal duplex sites, we assessed the enzymatic and signaling capabilities of RIG-I bound to RNA with no free ends. This RNA was created from two partially complementary strands that contain 5′ and 3′ regions incapable of base pairing. The two strands were annealed and ligated, creating a dumbbell-like structure that is characterized by a 14-bp duplex region flanked on each end by an 8 base, ssRNA loop region (*Figure 4A*).

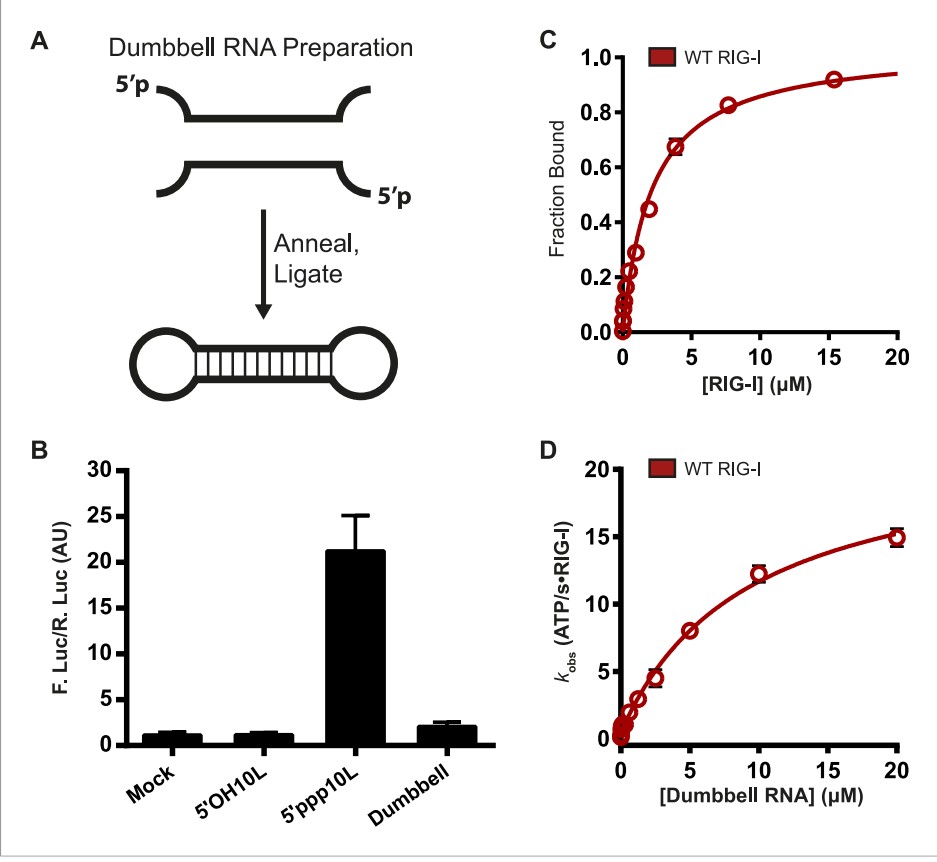

**Figure 4.** RIG-I cannot signal from internal sites. (**A**) Schematic representation of RNA dumbbell synthesis. (**B**) IFN-β induction of WT RIG-I in HEK-293T cells challenged with 5′OH10L, 5′ppp10L, and the RNA dumbbell. (**C**) Equilibrium binding of RIG-I to the RNA dumbbell. The fraction bound RNA dumbbell is plotted agaings protein concentration. Values are mean ± SD (n = 3). (**D**) Steady state ATP hydrolysis by wild type RIG-I stimulated with varying concentrations of the RNA dumbbell. Data were fit to the Brigg–Haldane equation yielding values of $k_{cat}$ = 21.6 ± 1.4 s$^{-1}$, K$_{M, RNA\ dumbell}$ = 9.3 ± 1.4 µM. Plotted values are mean ± SD (n = 3).

We first measured the ability of the RNA dumbbell to elicit an IFN-β response in cells. The dumbbell was unable to induce signaling by wild type RIG-I (*Figure 4B*, *Table 1*), demonstrating that RIG-I cannot signal from an exclusively internal binding site. Longer dumbbells (up to 50 bp) also fail to induce signaling, indicating that this effect is not dependent on duplex length (data not shown).

We next determined the affinity of RIG-I for a fluorescently-labeled, 14-bp RNA dumbbell using an electrophoretic mobility shift assay (see 'Materials and methods'). RIG-I exhibits weak affinity for the dumbbell, with a dissociation constant of $2.1 \pm 0.22$ µM (*Figure 4C*). By comparison, RIG-I binds blunt-ended RNA termini orders of magnitude more tightly, with dissociation constants ranging from about 200 pM to around 20 nM for 5′ triphosphorylated and 5′ hydroxyl ligands, respectively (*Vela et al., 2012*). Thus, while it is unable to signal from an internal duplex, RIG-I is still capable of binding such sites at very high concentrations.

We then evaluated the ATP binding and hydrolysis activity of RIG-I bound to the 14-bp dumbbell. We found that ATP binding by RIG-I on the dumbbell is qualitatively the same as that observed when bound to duplex RNA termini, and that the dumbbell is capable of stimulating ATP hydrolysis at a rate constant that is comparable to 5′ppp10L at saturating RNA concentrations (*Figure 4D*, *Table 1*). These data show that ATP binding and hydrolysis is not impaired when wild-type RIG-I is bound to internal duplexes. Taken together with the inability of RIG-I to signal on the dumbbell, we conclude that ATP binding and hydrolysis does not always lead to signaling. Because ATP binding and hydrolysis do not induce signaling by RIG-I on an internal duplex, we wondered if these activities might serve a second functional role.

## Nucleotide binding stimulates RNA dissociation by RIG-I

A number of DExD/H-box helicases, including the RLR MDA5, have been shown to use ATP binding and hydrolysis to promote dissociation from a bound ligand (*Liu et al., 2008*; *Berke and Modis, 2012*; *Berke et al., 2012*). Given the inability of RIG-I to signal on internal RNA duplexes, we hypothesized that this protein might also use ATPase activity to induce RNA dissociation in order to discriminate between optimal and non-productive RNA binding sites. A mechanism for differentiating target viral RNAs from diverse host RNAs would prevent undesirable induction of antiviral signaling and inflammation by endogenous RNA molecules. To investigate how ATP binding and hydrolysis affect RNA binding, we used fluorescence anisotropy to monitor RIG-I binding to a FAM-labeled 10-mer RNA hairpin alone or when bound to various nucleotides and nucleotide analogs. Equilibrium dissociation constants were determined in the presence of ADP, as well as the analogs ATPγS and ADP-AlF$_4$, which act as mimetics of ATP and the transition state of catalytic hydrolysis respectively (*Figure 5—figure supplement 1*). To ensure that ATPγS recapitulates interactions formed between RIG-I and ATP, we also determined RNA binding constants for Walker A mutants in the presence of ATP. WT RIG-I and Walker A mutant RIG-I exhibited significant losses in RNA binding affinity in the presence of all nucleotides tested. The most substantial changes in RNA $K_d$ values were observed in the presence of the ATP analog ATPγS, although interactions with both the transition state analog ADP-AlF$_4$ and ADP also impaired RNA binding by RIG-I (*Figure 5A*, *Table 1*). Further, ATP and ATPγS caused nearly identical losses in RNA affinity for both K270A and K270R RIG-I, indicating that ATPγS is a *bona fide* ATP mimetic for this system (*Table 1*). These experiments show that affinity of RIG-I for RNA is reduced by up to threefold in the presence of nucleotide.

While the equilibrium binding data are consistent with a nucleotide-stimulated dissociation mechanism, we next wanted to directly measure RIG-I dissociation from RNA to determine whether the dissociation rate constant is altered by nucleotide binding. To this end, microscopic rate constants for the dissociation of RIG-I ($k_{off}^{RNA}$) from 5′ppp10L were determined either alone or in the presence of various nucleotides and nucleotide analogs. Briefly, $k_{off}^{RNA}$ values were measured using stopped flow fluorescence spectroscopy, with a pre-bound RIG-I:dsRNA-Cy3 complex rapidly mixed with an excess of unlabeled RNA, resulting in the release of RIG-I from Cy3-RNA and a concomitant decrease in fluorescence intensity (see 'Materials and methods' and *Figure 5B*).

We first measured the $k_{off}^{RNA}$ for RIG-I in the absence of nucleotide to establish a baseline rate constant for dissociation. The observed time courses exhibited biphasic behavior and were best fit to a double exponential equation (*Figure 5B*). RIG-I dissociates slowly from 5′ppp10L, with $k_{off}$ values of $27.8 \pm 0.3 \times 10^{-3}$ s$^{-1}$ and $4.0 \pm 0.2 \times 10^{-3}$ s$^{-1}$, respectively (*Figure 5B*, *Table 2*). We interpret the biphasic nature of dissociation as representing two conformational populations of RIG-I present when

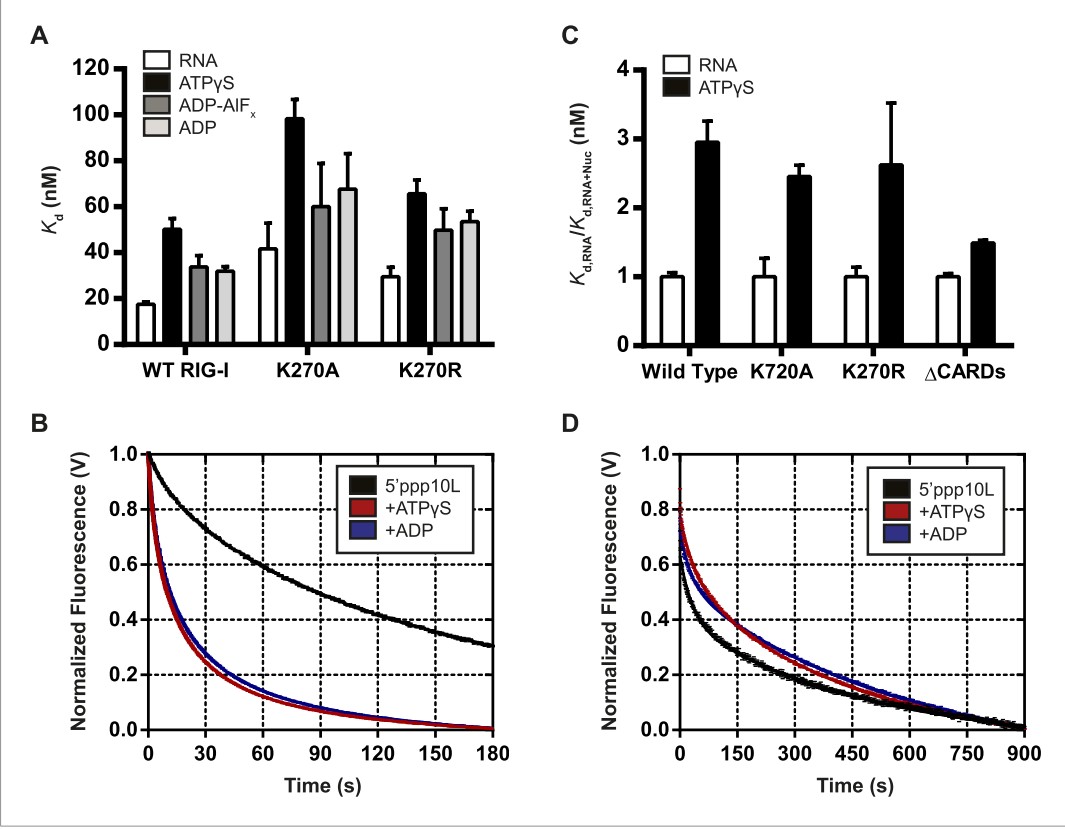

**Figure 5**. Nucleotide binding stimulates ligand dissociation in a CARDs-dependent manner. (**A**) Equilibrium dissociation constants for WT and mutant RIG-I binding to the fluorescent RNA hairpin 5'OH10L measured in the absence of nucleotide (denoted *RNA*), or in the presence of saturating ATPγS, ADP-AlF$_4$, or ADP. (**B**) $k_{off}$ traces for WT RIG-I. Four traces monitoring displacement of RIG-I from 5'ppp10L (black) were averaged and fit to a double exponential equation. Also shown are $k_{off}$ traces in the presence of 3 mM ATPγS (blue) and 3 mM ADP (red). (**C**) Relative dissociation constants for 5'OH10L binding by WT K270A, K270R, and ΔCARDs RIG-I proteins in the absence (*RNA*) or presence of saturating ATPγS normalized to *RNA*. (**D**) $k_{off}$ traces for RIG-I ΔCARDs performed as in (**B**) in the absence of nucleotide (black) or in the presence of ATPγS (red) or ADP (blue). Plotted values are mean ± SD (n = 3).

The following figure supplement is available for figure 5:

**Figure supplement 1**. Effect of nucleotide analogs on the ATPase activity of RIG-I.

bound to RNA, with most of the protein in a conformation corresponding to the slower dissociating species.

Having obtained basal $k_{off}^{RNA}$ values for RIG-I dissociation from 5'ppp10L, we next measured the $k_{off}^{RNA}$ in the presence of different nucleotides to evaluate the effects of ATP binding. As above, we rapidly mixed pre-formed RIG-I:Cy3-RNA complex with an excess of trap RNA, which now included saturating amounts of nucleotide. When we compared dissociation rate constants obtained in the presence of nucleotides to the basal RNA dissociation rate constant, we found that the presence of either ATPγS or ADP caused a five-fold increase in the rate of dissociation from 5'ppp10L (*Table 2*). In addition to the increased efficiency of dissociation, the relative contributions of the fast and slow populations to the ensemble dissociation rate constant also changed. While the faster rate constant comprised about 25% of the total change in fluorescence for basal $k_{off}^{RNA}$ measurements, it now comprised 40–45% of the amplitude change in the presence of nucleotide. This further suggests that a rapidly dissociating population of RIG-I is favored in the presence of nucleotide.

The $k_{off}^{RNA}$ values for the Walker A mutants were also determined in the absence and presence of nucleotide. Dissociation of the 5'ppp10L RNA was stimulated for both K270R and K270A in the

**Table 2**. RNA dissociation rate constants for WT and mutant RIG-I

| Protein | Nucleotide | $k_1$ (× $10^{-3}$ $s^{-1}$) | Amplitude | $k_2$ (× $10^{-3}$ $s^{-1}$) | Amplitude |
|---------|------------|------------|-----------|------------|-----------|
| WT RIG-I | None | 28 ± 13 | 25% | 4 ± 0.2 | 75% |
| | ATPγS | 187 ± 3 | 45% | 26 ± 0.4 | 55% |
| | ADP | 187 ± 3 | 40% | 25 ± 0.3 | 60% |
| K270A | None | 108 ± 1 | 80% | 5.3 ± 0.1 | 20% |
| | ATP | 327 ± 2 | 79% | 25 ± 0.3 | 21% |
| | ATPγS | 390 ± 3 | 75% | 35 ± 0.4 | 25% |
| K270R | None | 92.9 ± 1 | 53% | 16 ± 0.1 | 47% |
| | ATP | 227 ± 2 | 70% | 14 ± 0.1 | 30% |
| | ATPγS | 224 ± 1 | 81% | 16 ± 0.2 | 19% |
| ΔCARDs | None | 30.5 ± 0.5 | 34% | 2.1 ± 0.02 | 66% |
| | ATPγS | 24.0 ± 0.3 | 27% | 2.2 ± 0.01 | 73% |
| | ADP | 35.3 ± 0.5 | 23% | 1.7 ± 0.01 | 77% |

Rate constants for the dissociation of RIG-I proteins measured using stopped-flow fluorescence spectroscopy as described in 'Materials and methods'. Dissociation was measured from a 5'ppp10L hairpin RNA in all cases. The data were fit to a biphasic exponential decay equation and. Values represent mean ± SD (n = 3).

presence of ATP and ATPγS (*Table 2*) as with WT RIG-I. The ATPase dead mutants provided an accurate means to test the effect of ATP binding on $k_{\mathrm{off}}^{\mathrm{RNA}}$, although with WT RIG-I we also observed similar RNA dissociation rate constants in the presence of ATP and ATPγS (data now shown). As expected, these data demonstrate that ATP binding stimulates RIG-I dissociation from RNA, even in the absence of hydrolysis. We therefore conclude that one role of ATP binding is to challenge the RIG-I interaction with RNA and to promote dissociation. This activity may suppress RIG-I signaling from low affinity RNA binding sites, such as internal duplexes or non-triphosphorylated ends, liberating RIG-I to search for appropriate viral RNA targets.

## Nucleotide-induced RNA dissociation is linked to the CARDs

As a first step in establishing a physical mechanism for nucleotide-induced RNA dissociation, we examined whether the CARDs are important for this behavior. In a comparative binding analysis of the individual RIG-I domains, Vela et al. demonstrated that the CARDs antagonize RNA binding by helicase core and the CTD (*Vela et al., 2012*). Further, available structural data suggests that RIG-I bound to both RNA and the transition state analog ADP-AlF$_4$ cannot accommodate the CARDs:HEL2i interaction characteristic of the autorepressed state (*Kowalinski et al., 2011*; *Luo, 2014*; *Rawling and Pyle, 2014*). We therefore hypothesized that nucleotide binding and subsequent dissociation from RNA may be related to the antagonism between the CARDs and the HEL/CTD domains. To test this hypothesis, we measured the equilibrium RNA binding affinities and pre-steady state RNA dissociation rates of a RIG-I construct lacking the CARDs (ΔCARDs) in the presence of various nucleotides.

Compared to wild type and Walker A mutant constructs, where the presence of ATPγS resulted in an approximate threefold drop in affinity, ATPγS binding by the ΔCARDs construct lowered the affinity about 1.4-fold (*Figure 5C*, *Table 1*). Consistent with this observation, we detected no enhancement in the rate of RNA dissociation ($k_{\mathrm{off}}$) for ΔCARDs RIG-I in the presence of nucleotide (*Figure 5D*, *Table 2*). Taken together, these results suggest a significant role for the CARDs in mediating the enhanced RNA dissociation upon nucleotide binding.

## Discussion

In this report, we investigate the relationship between the ATPase and immune signaling activities of RIG-I when it is bound to dsRNA ligands of varying length and composition. We show that ATP binding, not hydrolysis, represents the minimal requirement for RIG-I signaling. Further, we

demonstrate that RIG-I cannot signal from a purely internal binding site, and that when RIG-I is bound to an internal duplex, ATP binding and hydrolysis promote dissociation from the RNA. Based on these results, we present a unified mechanism for ATP- dependent signaling and proof-reading by RIG-I.

## ATP binding, not hydrolysis, is required for signaling

To investigate the requirement for ATPase activity in RIG-I signaling, we produced RIG-I constructs with mutations at the key lysine residue of the highly conserved helicase motif I (Walker A motif). In RIG-I, this residue (K270) forms hydrogen bonds with the β and γ phosphates of ATP, and helps coordinate the $Mg^{2+}$ ion required for chemistry (*Jiang et al., 2011*; *Kowalinski et al., 2011*; *Luo et al., 2012b*). Hence, mutation of this residue is expected to significantly weaken ATP binding and abrogate hydrolysis. As expected, RIG-I K270 mutants exhibit no ATP hydrolysis activity. Yet surprisingly, the K270 mutants retain the ability to bind ATP and to engage immune signaling when challenged by an optimized RNA ligand, 5′ppp10L (*Kohlway et al., 2013*). Just as strikingly, the K270 mutants fail to bind ATP and they fail to induce signaling when bound to longer, suboptimal RNA ligands. Taken together, these data establish that ATP binding plays a critical role in the induction of signaling, and that ATP binding is controlled through simultaneous interactions between RIG-I and its RNA ligands, thereby providing a two-tier system for highly specific response to viral RNA.

Although the activity of RIG-I K270R/5′ppp10L complexes provided a useful tool for showing that ATP binding, rather than hydrolysis, is strictly required for signaling, it raises the question of why 5′ppp10L is so special. Why does this short, high-affinity triphosphorylated RNA ligand uniquely activate the RIG-I protein, and what is the significance of this observation? To answer this question, it is instructive to refer to the published literature on crystal structures of RIG-I complexes with RNA and ATP ligands. It has been reported that ATPase activity of wild type RIG-I is optimally activated by short RNA ligands (*Kato et al., 2008*; *Kohlway et al., 2013*), suggesting that these RNAs help 'prime' the protein for interactions with ATP. Many important ATP binding interfaces, such as the Q motif and motif VI (*Cordin et al., 2004*, *2006*; *Luo et al., 2011*; *Luo et al., 2012a*), are not perturbed by the K270 mutation, and therefore a strong ATP binding cleft can still be formed if it is reinforced by additional interactions with short RNA ligands. It is likely that the optimized 5′ppp10L ligand compensates for the deficiencies caused by removing lysine 270, pre-organizing the active site for binding to ATP. Intriguingly, many viral genomes are predicted to form short duplex panhandle structures with mismatches, bulges, and loops near the duplex terminus, thereby mimicking the conformational constraints imposed by 5′ppp10L (*Luo et al., 2012b*).

Structural data on human and duck RIG-I support a model in which interactions between RNA-bound RIG-I and an ATP molecule can lead to CARDs expulsion and subsequent signaling. Alignment of RIG-I in an apo conformation (PDB 4A2W) to structures of RIG-I bound only to RNA (PDB 4AY2) or bound to RNA and the ATP analog ADP-AlF4 (PDB 4A36) show that the CARDs of RIG-I can be accommodated in apo and RNA-bound states, but not when bound to both RNA and the ATP analog (*Jiang et al., 2011*; *Kowalinski et al., 2011*; *Luo et al., 2011*, *2012b*; *Kolakofsky et al., 2012*; *Luo, 2014*; *Rawling and Pyle, 2014*) (*Figure 6*). When bound to ADP-AlF4, RIG-I adopts a conformation that would force the CARDs and CTD domains into a steric collision, likely promoting CARDs ejection and RIG-I activation (*Figure 6*). Using the K270 mutant RIG-I and an optimized 5′ppp10L ligand, we have shown that the initial ATP binding event is sufficient to cause this conformational transition and activate RIG-I, even in the absence of ATP hydrolysis.

## RIG-I activation depends on end capping by the CTD

Viral RNAs that productively activate RIG-I contain blunt duplex ends bearing a 5′ triphosphate moiety. It is clear that both of these features allow RIG-I to distinguish between pathogenic RNAs and potential host targets, such as capped RNA, single stranded RNAs, miRNA, or other processed RNAs in which the 5′ triphosphate moiety has been removed. However, the biophysical basis for this discrimination is not clearly understood. Analysis of our functional data and available crystallographic information also explain why RIG-I requires a blunt RNA duplex terminus for signaling. A RIG-I signaling model that involves ATP-mediated domain compaction depends upon a steric collision between the CARDs and the RNA-bound CTD in order to promote CARD expulsion (*Figure 7*, Left). Disrupting the CTD:RNA interface is expected to disrupt signaling by preventing this steric clash, and preventing CARD release. The tight binding affinity between the CTD and optimized RNA ligands

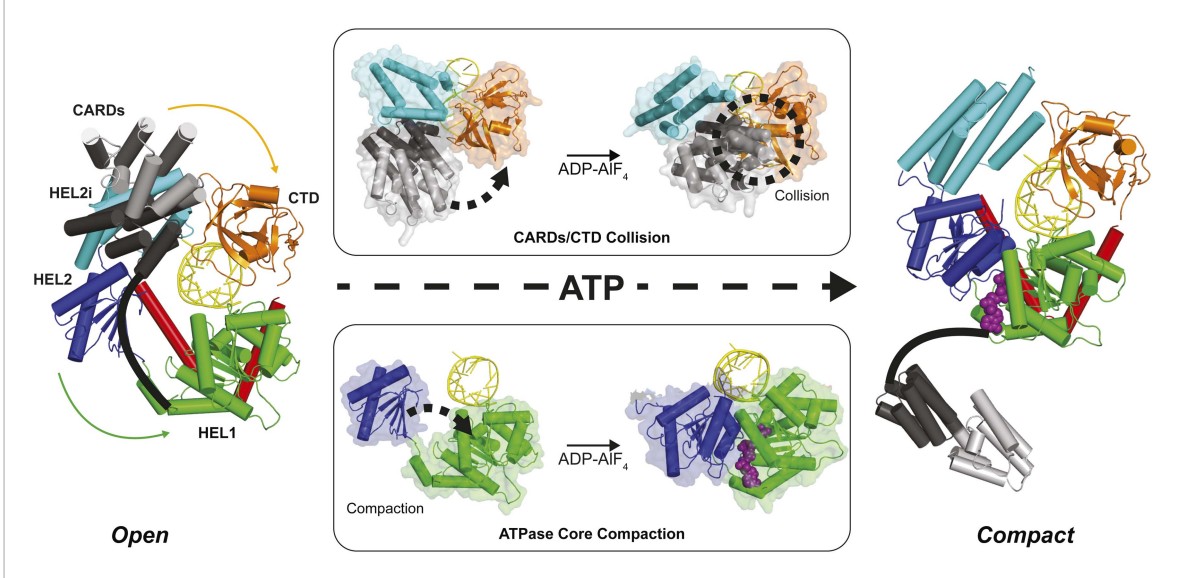

**Figure 6**. Structural description of a nucleotide-mediated collision between the CARDs and CTD of RIG-I. (Left) Structural model of RIG-I in an 'Open' conformation produced by docking the CARDs of full-length duck RIG-I (PDB: 4A2W) to human RIG-I bound to the hairpin 5'ppp8L (PDB: 4AY2) by alignment of the HEL2i domains. (Right) Structural model of RIG-I in a 'Compact' conformation produced by alignment of the CARDs, CTD and RNA from the 'Open' model to the HEL1, HEL2, HEL2i and Pincer domains of RIG-I bound to ADP-AlF$_4$ (PDB: 4A36). The middle panels show van der Waals surface representations demonstrating motions in the HEL2i/CARDs relative to the CTD that result in a steric collision (Top) and motions in the HEL1 and HEL2 domains of the ATPase core (Bottom) that take place upon nucleotide-mediated compaction. CARDs are shown black/gray; HEL1, green; HEL2, blue; HEL2i, cyan; Pincer, red; CTD, orange; RNA, yellow; and ADP-AlF$_4$ as purple spheres.

derives from a complex network of specific interactions with the 5′ triphosphate moiety and stacking interactions with the terminal base pair of an RNA duplex (*Lu et al., 2010*; *Jiang et al., 2011*; *Kowalinski et al., 2011*; *Luo et al., 2011*, *2012b*). By contrast, the CTD of RIG-I molecules bound at internal sites is effectively disengaged, and it is likely that all interactions between RIG-I and internal sites occur exclusively through contacts involving the helicase domain (*Vela et al., 2012*). In such cases, the CTD will not provide a hard barrier for colliding with the CARDS, and these will fail to disengage upon ATP binding and domain compaction (*Figure 7*, Right). We tested this model by creating the dumbbell ligand as a model for internal binding by RIG-I, and we observed that wild type RIG-I binds internal duplexes much more weakly than duplex termini. Importantly, RIG-I binds the dumbbell RNA with similar affinity as the isolated helicase domain binds to optimal, terminal duplexes (*Vela et al., 2012*), consistent with the fact that the CTD is disengaged when RIG-I binds internal sites. Nonetheless, the dumbbell ligand stimulates robust ATPase activity by wild-type RIG-I, indicating that the protein is capable of both binding and hydrolyzing ATP at internal duplexes, but it is incapable of signaling. While the ATP hydrolysis activity is likely to serve an important function (vide infra), the failure of this ligand to induce signaling is consistent with a compaction and collision model that requires a tightly anchored CTD. We therefore conclude that at internal sites, RIG-I binds and hydrolyzes ATP, but the signaling domains remain in their auto-repressed conformation.

The observation that monomeric RIG-I is incapable of signaling when bound to an internal duplex site does not preclude a possible role for additional RIG-I molecules which might load next to RIG-I that is bound at a blunt duplex terminus. Such a scenario could conceivably occur at very high concentrations of RIG-I. The proposed 'priming' behavior observed for the 5′ppp10L hairpin could potentially be recapitulated by tightly stacking additional copies of the protein against one another (*Figure 3—figure supplement 1*), thereby mimicking conformational constraints imposed by the short RNA and enhancing the signaling efficiency of the end-bound RIG-I. For example, we observe a significant and reproducible jump in RIG-I signaling activity when challenged by a 5′ppp50L hairpin compared to the 5′ppp30L hairpin. Even in this case, the requirement for CTD end-capping and the extremely weak affinity for internal duplex sites underscore the importance of an available duplex terminus for activating RIG-I signaling.

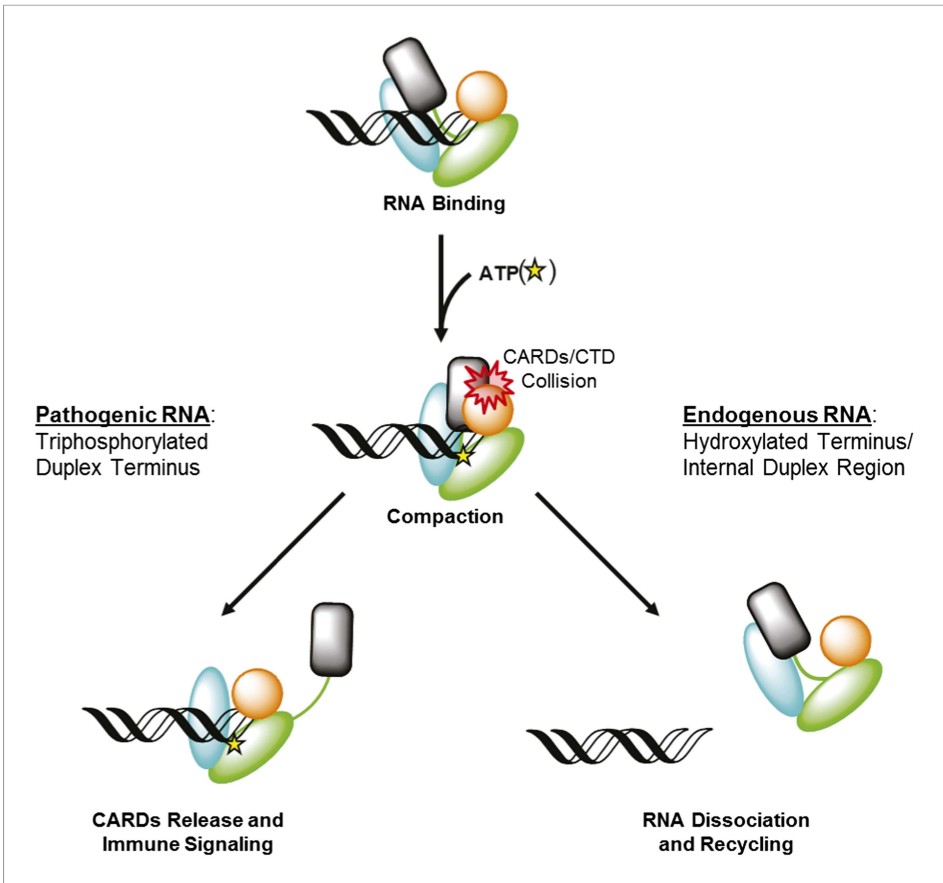

**Figure 7**. RIG-I uses ATP-mediated compaction as signaling trigger and a proofreading mechanism. RIG-I surveys the cytoplasm in an auto-repressed conformation, with the signaling CARDs (black) sequestered through an interaction with the HEL2i domain (cyan). Duplex RNA binding by RIG-I induces the formation of an active site pocket in the helicase core (green), which allows the protein to bind ATP (yellow star). ATP binding causes the domains of RIG-I to compact, bringing the CARDs into a steric collision with the CTD (orange). If the CTD is engaged in a high-affinity interaction with a triphosphorylated duplex terminus (as in viral RNA), this collision disengages the CARDS from the HEL2i domain, making them available for immune signaling (left). If the CTD is engaged in a low-affinity interaction with a hydroxylated terminus or internal duplex binding site, the CARDs/CTD collision destabilizes RNA binding by RIG-I, resulting in dissociation and recycling (right).

## RIG-I uses ATP to recycle from non-productive RNAs

Experiments performed using the dumbbell RNA demonstrate the importance of end binding in RIG-I activation, but they also show that RIG-I is capable of binding to sub-optimal duplexes, and that these RNA molecules stimulate non-productive ATPase activity. To deal with this issue, RIG-I has evolved a strategy for 'rejecting' non-pathogenic ligands by dissociating rapidly from these RNAs. RIG-I bound to ATP exhibits reduced equilibrium binding affinity for RNA, and rapid RNA dissociation. Furthermore, the observed enhancement in RNA dissociation kinetics is dependent on the CARDs of RIG-I, which suggests that a collision between the CARDs and the CTD further destabilizes RNA binding by RIG-I. Intriguingly, enhanced dissociation kinetics are observed for all RNAs tested, including those containing a 5′ triphosphorylated duplex terminus. Therefore, ATP-mediated turnover is not specific for a particular RNA ligand. Rather, it challenges and tests the RIG-I:RNA interaction for all ligands, and results in CARDs expulsion only when the CTD is rigidly constrained by high-affinity binding to the terminus of triphosphorylated viral RNAs (*Figure 7*).

Using the energy from ATP hydrolysis as a recycling and proof-reading mechanism has been previously demonstrated for RNA helicases belonging to the DEAD-box family of proteins, which is the helicase subgroup that is phylogenetically most closely related to RIG-I (*Jankowsky and Fairman, 2007*;

*Liu et al., 2008*; *Jankowsky, 2011*; *Luo et al., 2012a*). Indeed, a number of DEAD-box proteins are capable of performing duplex unwinding in the presence of a non-hydrolyzable ATP mimetic ADP-BeF$_3$ (*Liu et al., 2008*). The model presented here mirrors these findings; in both RIG-I and DEAD-box proteins, ATP binding is required for a particular mechanical activity, while ATP hydrolysis is only required to 'reset' the protein, enabling it to search for another target or to engage in another round of activation. Thus, ATP binding represents the deterministic step in RIG-I activation, while hydrolysis helps the protein to efficiently sample potential RNA ligands, and reject those that cannot activate immune signaling.

## RIG-I integrates RNA and ATP binding to function as an advanced biosensor

RIG-I integrates the input of ATP and RNA binding to select viral RNA over endogenous RNA, and subsequently initiate signaling. Endogenous RNAs, such as tRNAs, inverted repeat sequence elements, and pre-and mature miRNAs, are abundant in the cytoplasm and contain structured duplex regions to which RIG-I can readily bind (*Athanasiadis et al., 2004*; *Wang and Carmichael, 2004*; *Watanabe et al., 2008*; *Wang and Leung, 2009*; *Chiang et al., 2010*; *Kirchner and Ignatova, 2015*). The high cellular concentrations of these RNAs means RIG-I will bind at these sites despite its relatively weak affinity for RNA that is not triphosphorylated. Importantly, RIG-I can bind these RNAs without activating signaling, due to the proofreading mechanism imparted by ATP hydrolysis. Thus, RIG-I is thus able to differentiate between non-pathogenic RNAs and viral RNAs using a finely tuned proofreading mechanism that is essential for preventing aberrant immune responses.

# Materials and methods

## RNA hairpin preparation

Oligoribonucleotides were synthesized on an automated MerMade synthesizer (BioAutomation, Irving, TX, United States) using standard phosphoramidite chemistry. The oligonucleotides were deprotected and gel purified as previously described (*Wincott et al., 1995*). Fluorophore-labeled RNA hairpin used in stopped flow fluorescence spectroscopy was generated using amino-modified RNA (on a U in the UUCG tetraloop), which is synthesized using a 3′ amino modifier with a C3 linker (Glen Research, Sterling, VA, United States). Cy3 Mono NHS ester (GE Healthcare, Little Chalfont, United Kingdom) was conjugated to the modified oligonucleotides as per the manufacturer's instructions. Cy3-labeled and unlabeled RNA were separated on a 20% denaturing polyacrylamide gel, and purified by gel extraction.

Triphosphorylated RNA hairpins were prepared by in vitro transcription by T7 RNA polymerase using DNA oligomers containing 3′ 2′-O-methyl modifications as templates. After a 5 hr transcription at 37°C, the resulting transcripts were purified by gel extraction from a denaturing polyacrylamide gel (12–20%, depending on RNA transcript length). To assess purity, the purified RNA hairpins were then run on a 15% native polyacrylamide gel for 60 min at 100 V, stained in a 1:10,000 dilution of SYBR Gold Nucleic Acid Gel Stain (Life Technologies, Carlsbad, CA, United States) in 0.5× TBE Running Buffer (Life Technologies) and imaged on a Typhoon FLA 9500 biomolecular imager (GE) (*Figure 3—figure supplement 2*). All RNA sequences are listed in *Supplementary file 1*.

## RNA dumbbell preparation

To create an RNA ligand without termini, an RNA dumbbell consisting of a 14 base pair stem region and two, 8 nucleotide single stranded loops was made. The synthesis of an RNA dumbbell requires two RNA strands, where two ligation events occur to yield an RNA without termini. The 14 base pair dumbbell was converted into two precursor RNA strands by placing nicks in the single strand loop regions to allow for straightforward ligation by T4 RNA ligase 1 (for sequences see *Supplementary file 1*). The two RNA strands containing 5′ monophosphates were synthesized using standard phosphoramidite chemistry as described above, with 5′ monophosphorylation performed on the synthesizer using Chemical Phosphorylation Reagent (Glen Research, Sterling, VA, United States). After deprotection and gel purification, the two strands (30 µM each) were annealed by rapid heating to 95°C followed by slow cooling to 30°C in annealing buffer (100 mM MOPS pH 7.5, 10 mM DTT). The duplexed RNA was then ligated using T4 RNA ligase 1 (New England Biolabs, Ipswich, MA, United States) in ligation buffer (100 mM MOPS pH 7.5, 10 mM DTT, 10% DMSO, 15 mM MgCl$_2$, 0.6 mM ATP) for 12–15 hr at 16°C.

The ligated RNA dumbbell was then purified to 95–98% purity (see below, *Figure 3—figure supplement 2*) using denaturing polyacrylamide gel electrophoresis and gel extraction. For Cy3-labeled dumbbell, the same synthetic methods were employed as described above. One of the precursor RNA strands contains an amino-modified U in a single strand loop region, which is synthesized as described above for the amino-modified hairpin. After ligation and initial purification, Cy3 Mono NHS ester (GE Healthcare) was conjugated to the modified dumbbell. The Cy3-labeled RNA dumbbell was then isolated using denaturing polyacrylamide gel electrophoresis and gel extraction.

To assess the purity of the RNA dumbbell, a 5′ end labeling reaction using [γ-$^{32}$P] ATP (Perkin Elmer, Waltham, MA, United States) and T4 polynucleotide kinase (New England Biolabs) was performed (*Figure 3—figure supplement 2*). If both ligation events occurred and the dumbbell was formed, the labeling reaction would be unsuccessful as there are no free 5′ ends. In contrast, any impurities (i.e., hairpin or duplex RNA resulting from 1 or no ligation events, respectively) in the sample would be 5′ end labeled. First, samples were treated with Antarctic Phosphatase (New England Biolabs) as per the manufacturer's instructions to remove any 5′ monophosphates. After heat inactivation, 15 pmol of the Antarctic phosphatase treated RNA was incubated with T4 PNK and excess [γ-$^{32}$P] ATP at 37°C for 1 hr. To assess purity of the dumbbell in a quantifiable manner, the incorporation efficiency of [γ-$^{32}$P] was determined using a filter spotting technique. Briefly, 1 μl of the reaction was spotted in duplicate on Whatman DE81 filter papers. One filter was washed by immersion in 25 ml 0.5 M sodium phosphate pH 7.0 buffer for 5 min (four times). After drying, the washed and unwashed filters were exposed on a phosphorimaging screen and scanned using the Typhoon phosphorimager (GE Healthcare). The percent incorporation was then determined by dividing quantified intensity for the washed filter by quantified intensity for the unwashed filter. Using the specific activity of [γ-$^{32}$P] ATP, the pmol of RNA labeled was also calculated.

## HEK 293T cell culture and IFN-β induction assays

Vector pUNO-hRIG-I for constitutive WT RIG-I expression in mammalian cells was purchased from Invivogen. Mutations were introduced into the parent plasmid using appropriate primers K270A forward: 5′-taagcagtgaaacaaaggttgctccacaacctgtaggagcac-3′ and K270A reverse: 5′-gtgctcctacaggttgtggagcaacctttgtttcactgctta-3′ or K270R forward: 5′-gcagtgaaacaaaggttcttccacaacctgtaggagc-3′ and K270R reverse: 5′-gctcctacaggttgtggaagaacctttgtttcactgc-3′ and PfuUltra Hotstart PCR Master Mix (Agilent, Santa Clara, CA, United States) per the manufacturer's protocol. Mutagenesis was validated by sequencing.

Cell based experiments were conducted in HEK 293T cells because they do not express endogenous RIG-I (proteinatlas.org). Cells were grown and maintained in 15 cm dishes containing Dulbecco's Modified Eagle Medium (DMEM; Life Technologies) supplemented with 10% heat-inactivated fetal bovine serum (Hyclone, GE Healthcare) and Non-Essential Amino Acids (Life Technologies). IFN-β induction assays were conducted in 6-well format. Briefly, 2.5 ml of cells at 100,000 cells/ml were seeded in each well of a tissue culture treated 6-well plate. After 24 hr, each well of cells was transfected with the indicated amount of WT or mutant pUNO-hRIG-I, 30 ng pRL-TK constitutive Renilla luciferase reporter plasmid (Promega, Madison, WI, United States), and 750 ng of an IFN-β/FireflyLuc reporter plasmid using the Lipofectamine 2000 transfection reagent (Life Technologies) per the manufacturer's protocol. Protein expression was allowed to proceed for 24 hr, at which point the cells were challenged by transfection of 2.5 μg of the indicated dsRNA, also using the Lipofectamine 2000 reagent. After 12 hr, cells were harvested for luminescence analysis.

To assess IFN-β induction using a dual luciferase assay, cells were harvested and lysed as follows: Growth media was aspirated from each well, and 200 μl of passive lysis buffer (Promega) was added. Lysis proceeded for 15 min at room temperature. The lysates were transferred to a 96-well PCR plate (Eppendorf, Hamburg, Germany) and clarified by centrifugation. Next, 20 μl samples of the supernatant were transferred to a 96-well assay plate for analysis using the Dual-Luciferase Reporter Assay System (Promega). Luminescence was measured using a Biotek Synergy H1 plate reader (Biotek, Winooski, VT, United States). The resulting Firefly luciferase activity (i.e., the induction of IFN-β) was normalized to the activity of the constitutively expressed Renilla luciferase to account for differences in confluency, viability and transfection efficiency across sample wells.

## Western blot analysis

For Western blot analysis, 20 μl of the appropriate HEK 293T cell supernatant was removed and combined with 5 μl SDS-PAGE loading dye. 15 μl samples of this mixture were run on a 4–20% Mini-PROTEAN TGX

gel (Bio-Rad, Hercules, CA, United States). Proteins were transferred to a PVDF membrane at 100 V for 60 min. Blocking was performed at 4°C for 4 hr in 5% BSA dissolved in TBS buffer. The membrane was then washed in TBST buffer 3× for 5 min each at 23°C. The membrane was next incubated in 3% BSA/TBS solution containing primary αRIG-I antibody (Sigma, St. Louis, MO, United States) at a 1:1000 dilution overnight at 4°C. The following morning, the membrane was washed in TBST 3× for 5 min each at 23°C. The membrane was then incubated in 3% BSA/TBS solution containing secondary αRabbit:HRP antibody at a 1:10,000 dilution for 30 min at 23°C. The membrane was washed in TBST 3× for 10 min each at 23°C, then treated with SuperSignal West Pico Chemiluminescent Substrate (Thermo Scientific, Waltham, MA, United States) per the manufacturer's protocol. Chemiluminescence was visualized by film.

## Protein purification

For expression, plasmids were transformed into Rosetta II (DE3) *Escherichia coli* cells (Novagen, Madison, WI, United States) and grown in LB media supplemented with 50 mM Potassium Phosphate pH 7.4 and 1% glycerol. Expression was induced by the addition of isopropyl-β-D-thiogalactopyranoside to a final concentration of 0.5 mM. Cells were grown for 24 hr at 16°C, then harvested by centrifugation, resuspended in lysis buffer (20 mM Phosphate pH 7.4, 500 mM NaCl, 10% glycerol, 5 mM β-mercaptoethanol [βME]) to a final volume of 50 ml, and frozen at −80°C. For lysis, frozen pellets were thawed at room temperature, then resuspended in an additional 200 ml lysis buffer per 4L pellet. Cells were lysed by passage through a microfluidizer at 15,000 psi, and the lysate was clarified by ultracentrifugation at 100,000×$g$ for 30 min. Soluble lysate was incubated on 2.5 ml Ni-NTA beads (Qiagen, Valencia, CA, United States), washed with lysis buffer containing an additional 40 mM imidazole, then eluted in Ni elution buffer (25 mM HEPES pH 8.0, 150 mM NaCl, 220 mM Imidazole, 10% glycerol, 5 mM βME). Eluted protein was bound to a HiTrap Heparin HP column (GE Healthcare), washed in buffer containing 150 mM NaCl, and eluted stepwise at 0.65 M NaCl. The SUMO tag was then removed by incubation with SUMO protease for 2 hr at 4°C. Finally, monomeric protein was collected by passage over a HiPrep 16/60 Superdex 200 column (GE Healthcare) in gel filtration buffer (25 mM MOPS pH 7.4, 300 mM NaCl, 5% glycerol, 5 mM βME). Peak fractions were concentrated to 10–20 μM using a centrifugal concentrator with a 50 kD molecular weight cutoff (Millipore, Billerica, MA, United States). Concentrations were determined spectrophotometrically using an extinction coefficient of $\varepsilon = 99,700$ $M^{-1}$ $cm^{-1}$ at $\lambda = 280$ nm. Protein preparations were aliquoted, flash frozen using liquid nitrogen, and stored at −80°C.

## NADH coupled ATPase activity assay

RIG-I ATPase activity was measured using an established absorbance-based coupled assay system. The RIG-I protein of interest was diluted into ATPase assay buffer (25 mM MOPS pH 7.4, 150 mM NaCl, 5 mM DTT) to a final concentration of 10 nM for $K_{M, RNA}$ experiments in the presence of a coupled assay mix (1 mM NADH, 100 U/ml lactic dehydrogenase, 500 U/ml pyruvate kinase, 2.5 mM phosphoenol pyruvic acid).

For $K_{M, RNA}$ experiments, the RNA of interest was diluted into ME buffer (25 mM MES pH 6.0, 0.5 mM EDTA) over a 12-pt concentration series and added to the protein/NADH sample mix resulting in RNA concentrations varying from approximately 0.5 nM–500 nM. Samples were incubated for at least 2 hr at room temperature. The reaction was initiated by the addition of 5 mM ATP/5 mM $MgCl_2$ to all wells.

The rate of ATP hydrolysis was determined indirectly by monitoring the conversion of NADH to $NAD^+$ which results in a loss of sample absorbance at 340 nM. The assay was performed in 96-well format and absorbance was measured over a 10 min time course using a Biotek Synergy H1 Plate Reader (Biotek). Mean velocities were extrapolated for each time course and plotted as a function of either ATP or RNA concentration. These data were then fit to the quadratic solution of the Briggs-Haldane equation:

$$y = y_0 + (amp) * \frac{x + p + K_M - \sqrt{(x + p + K_M)^2 - 4xp}}{2p}, \tag{1}$$

where $y_0$ = basal activity, defined as background catalytic velocity observed in the absence of RIG-I, amp = $v_{max} - y_0 = k_{cat}$, x = total ATP or RNA concentration, p = total protein concentration, and $K_M$ is the Michaelis constant for the variable substrate.

## Equilibrium fluorescent RNA binding assays

The fluorescent RNA hairpin used in binding experiments (TriLink Biotech, San Diego, CA, United States) contained a 10 base pair duplex capped by a tetraloop with the sequence GGACGUACGUUU (6-FAM)CGACGUACGUCC and included an internal fluorescent modification, carboxyfluorescein. Binding assays were carried out in 384-well plate format. Briefly, dsRNA was diluted into binding buffer (25 mM MOPS pH 7.4, 150 mM NaCl, 5 mM DTT, 2 mM MgCl, 0.01% Triton X-100) to a concentration of 2 nM. The RIG-I protein construct of interest was then diluted into binding buffer over a 12-pt series of concentrations and mixed 1:1 with RNA samples (final RNA concentration of 2 nM) to a volume of 20 μl. Final RIG-I concentrations varyied from 1.5 nM to 1500 nM. Samples were equilibrated at room temperature for 2 hr. Fluorescence polarization was measured using a Biotek Synergy H1 plate reader (Biotek). Samples were excited through a bandpass filter at 485/20 nM and fluorescence emission was measured through a bandpass filter at 528/20 nM. Polarization was calculated using the following *Equation 2*:

$$P = \frac{I_{||} - G*I_{\perp}}{I_{||} + G*I_{\perp}},$$ (2)

where $I_{||}$ is the intensity of the fluorescent light parallel to the plane of excitation, $I_{\perp}$ is the intensity of fluorescent light perpendicular to the plane of excitation, and G is an empirically determined correction factor accounting for instrumental bias toward the detection of horizontally polarized light; in this case G = 0.87.

An individual experiment consisted of two replicates of each protein concentration for which polarization measurements were taken three times, yielding six values for each condition. The mean polarization values were then plotted against protein concentration and fit to a one-site total binding *Equation 3*:

$$y = y_0 + \frac{y_{max}*x}{K_d + x},$$ (3)

where $y_0$ represents the polarization value when [enzyme] = 0 nM, $y_{max}$ represents the polarization achieved at a saturating enzyme concentration, and $K_d$ is the dissociation constant. Three experiments were performed for each RIG-I construct, with each reported $K_d$ value representing the mean and standard deviation across these experiments.

For equilibrium nucleotide interference analysis, binding experiments were conducted identically as above, except that indicated nucleotides were present in the RNA/binding buffer mixture such that they achieved a final concentration 2 mM.

## RNA dissociation rate constant measurements

To obtain RNA dissociation rate constants ($k_{off}^{RNA}$) for RIG-I, stopped-flow fluorescence spectroscopy was employed. Stopped-flow experiments were performed in buffer A (25 mM HEPES pH 7.4, 150 mM NaCl, 2 mM DTT, 0.1 mg/ml BSA) at 24°C using a Kintek Auto-SF stopped-flow instrument (Kintek, Austin, TX, United States) supplied with a 150 W xenon arc lamp. For detection, the Cy3-labeled hairpin RNA was excited at 515 nm and the fluorescence emission was monitored at ≥570 nm using a 570 bandpass filter (Newport Corporation, Irvine, CA, United States). Briefly, RIG-I was pre-incubated with Cy3-labeled hairpin RNA in equimolar amounts (400 nM) at room temperature for 2–6 hr to form a protein-RNA complex. The protein-Cy3 RNA complex was then rapidly mixed with a 100-fold excess of unlabeled hairpin RNA for a specified period of time in which 2000 points were collected. For experiments with nucleotide, adenosine 5′-triphosphate (ATP) (Sigma), adenosine 5′-diphosphate (ADP) (Sigma) or adenosine 5′-[γ-thio]triphosphate (ATPγS) (Sigma) and MgCl₂ were included with the trap RNA and rapidly mixed with the protein-Cy3 RNA complex. The average fluorescence measurements (4–6 traces) for each condition were then used in data analysis.

Data was fit using non-linear regression to a single *Equation 4* or double exponential *Equation 5* using GraFit 5,

$$y = A_0 e^{-kt} + \text{offset},$$ (4)

$$y = A_{0(1)} e_1^{-kt} + A_{0(2)} e_2^{-kt} + \text{offset},$$ (5)

where A is the amplitude, $k$ is the rate constant, t is the reaction time (s), and the offset is the fluorescence value (V) of free Cy3-RNA.

## MANT-ATP binding

Using stopped flow fluorescence spectroscopy, RIG-I binding to nucleotide was measured via Förster resonance energy transfer from RIG-I ($\lambda_{ex}$ 290) to the MANT-ATP (Invitrogen Life Technologies) ($\lambda_{em} >$ 400). RIG-I (1 μM) bound to the specified RNA hairpin (1 μM) in buffer A (see above) was mixed with varying concentrations of mant-nucleotide (10–160 μM) under pseudo-first order conditions (>4× [RIG-I]) to measure the observed association rate constant. Upon binding ($\lambda_{ex}$ 290), the change in fluorescence was monitored through a 400 nm long-pass filter using the Kintek Auto-SF stopped flow (Kintek), and the resulting traces were fit to a double exponential equation (*Equation 5*, above). The $k_{obs}$ ($k_1$) corresponding to the initial binding event (obtained in triplicate) is then plotted vs (mant-nucleotide) and fit to a linear *Equation 6* using GraFit 5,

$$y = mx + b, \tag{6}$$

where the intercept y is $k_{off}$ and the slope m is $k_{on}$. A calculated $K_d$ can then be obtained ($k_{off}/k_{on}$) using the experimentally derived $k_{off}$ and $k_{on}$ values (*Figure 2—figure supplement 1D–F*).

## Electrophoretic mobility shift assays

For EMSAs performed to determine the equilibrium dissociation constants of WT RIG-I for the RNA dumbbell, reactions (20 μl) containing 10 nM Cy3-labeled RNA dumbbell and varying concentrations of protein (15 nM–15.4 μM) were incubated at room temperature for 30 min in RNA binding buffer (25 mM HEPES pH 7.4, 150 mM NaCl, 2 mM DTT, 0.1 mg/ml BSA). A portion of the reactions were loaded onto a precast 6% native polyacrylamide gel (Invitrogen) and run in 0.5× TBE at 100 V for 55 min at 4°C. The gels were imaged using the Typhoon FLA 9500 biomolecular imager (GE Healthcare) and analyzed using ImageQuant software (GE Healthcare). The fraction bound RNA was quantified for each protein concentration and plotted vs protein concentration. The data was fit to a one-site specific binding *Equation 7* using GraphPad Prism,

$$y = B_{max} * X / (K_d + X), \tag{7}$$

where y is specific binding, $B_{max}$ is the maximal fraction bound, X is the concentration of the Cy3-dumbbell RNA, and $K_d$ is the dissociation constant.

For EMSAs used to evaluate protein multimerization, wild type RIG-I was incubated with 100 nM RNA ligand for 30 min at room temperature in binding buffer (25 mM MOPS pH 7.4, 150 mM NaCl, 5 mM DTT, 2% glycerol, 0.1 mg/ml BSA) at concentrations ranging from 100 nM to 1 μM. After incubation, 1 μl of 20× loading dye (binding buffer with 0.4% wt/vol bromophenol blue and 0.4% wt/vol xylene cyanol) was added to each reaction, and 5 μl of each sample was run on a 6% DNA Retardation Gel (Life Technologies) for 60 min at 100 V. To visualize the RNA, the gel was stained in a 1:10,000 dilution of SYBR Gold Nucleic Acid Gel Stain (Life Technologies) in 0.5× TBE Running Buffer (Life Technologies) and imaged on a Typhoon FLA 9500 biomolecular imager (GE).

## Malachite green ATP hydrolysis assays

To screen non-hydrolyzable ATP analogs of RIG-I, the ability of these analogs to inhibit RIG-I ATPase activity was tested. The rates of ATP hydrolysis for RIG-I in the presence of ATP analogs (ATPγS, ADPAlF$_4$, AMPPNP) and ADP were determined using an established malachite green assay. RIG-I (50 nM) was first pre-incubated with an excess of 5′ppp10L (500 nM) for 2.5 hr at room temperature in buffer A (25 mM HEPES pH 7.4, 150 mM NaCl, 2 mM DTT) with 0.1 mg/ml BSA. The reactions were then initiated by the addition of a 1:1 ATP:MgCl$_2$ complex (1 mM each) in the presence and absence of ATP analog (4 mM). For initial velocity measurements ($v_0$), aliquots of the reaction were quenched at six time points between 15 s and 10 min. The reactions were quenched by the addition of 5× quench buffer (250 mM EDTA) for a final concentration of 50 mM EDTA. Malachite green reagent was added (9:1 malachite green: reaction volume) and allowed to age for 30 min at room temperature. The Abs.$_{650}$ was then measured using a Biotek Synergy 2 plate reader (Biotek). The rate of ATP hydrolysis for each reaction was determined by calculating the $k_{obs}$ ($k_{obs} = v_0/[E_{tot}]$). The relative rates

of ATP hydrolysis in the presence of the ATP analogs (as compared to the rate of hydrolysis without analog) reflect inhibition of RIG-I ATPase activity by the analogs.

The malachite green assay was used to determine the steady state kinetic parameters ($k_{cat}$ and $K_M$) of MANT-ATP hydrolysis for RIG-I. RIG-I (50 nM) was first pre-incubated with a saturating concentration of 5′ppp10L (500 nM) for 2.5 hr at room temperature in buffer A (25 mM HEPES pH 7.4, 150 mM NaCl, 2 mM DTT) with 0.1 mg/ml BSA. The reactions were then initiated by the addition of a 1:1 ATP:MgCl$_2$ complex. For initial velocity measurements, aliquots of the reaction were quenched at six time points between 15 s and 10 min. The reactions were quenched by the addition of 5× quench buffer (250 mM EDTA) for a final concentration of 50 mM EDTA. Malachite green reagent was added (9:1 malachite green: reaction volume) and allowed to age for 30 min at room temperature. The Abs.$_{650}$ was then measured using a Synergy 2 plate reader (BioTek). The steady state kinetic parameters, $k_{cat}$ and $K_M$, were obtained by determining the initial velocity ($v_0$) as a function of ATP concentration, and fitting the data to Michaelis-Menten equation (below) using non-linear regression using GraphPad Prism,

$$k_{obs} = k_{cat}\{[S]/(K_M + [S])\},$$

where $k_{obs} = v_0/[E_{tot}]$, and $E_{tot}$ is the total protein concentration.

## Acknowledgements

The authors thank Srinivas Somarowthu for editorial contributions. MEF is a Postdoctoral Associate and AMP is an Investigator with the Howard Hughes Medical Institute.

## Additional information

### Funding

| Funder | Grant reference | Author |
| --- | --- | --- |
| Howard Hughes Medical Institute (HHMI) | N/A | Megan E Fitzgerald, Anna Marie Pyle |

The funder had no role in study design, data collection and interpretation, or the decision to submit the work for publication.

### Author contributions

DCR, MEF, Conception and design, Acquisition of data, Analysis and interpretation of data, Drafting or revising the article; AMP, Conception and design, Analysis and interpretation of data, Drafting or revising the article

## Additional files

### Supplementary file

• Supplementary file 1. Sequences of RNA ligands used in this study. All RNA ligands and the sites of pertinent modifications are shown with the corresponding shorthand designation for each RNA.

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
