## [Decision Letter]

Thank you for submitting your work entitled “Establishing the role of ATP for the function of the RIG-I innate immune sensor” for peer review at *eLife*. Your submission has been favorably evaluated by John Kuriyan (Senior Editor), a Reviewing Editor, and three reviewers.

The reviewers have discussed the reviews with one another and the Reviewing Editor has drafted this decision to help you prepare a revised submission.

RIG-I recognizes viral RNA in the cytoplasm and initiates a host defense response, yet host RNA does not initiate a signaling response. Major questions remain in relation to how RIG-I distinguishes endogenous RNA from viral RNA. In this manuscript, Rawling et al. investigated the importance of ATP binding step in RIG-I activation, using biochemical and cell-based approaches. They examine RNA binding, ATP binding, ATP hydrolysis and IFN-β activation by RIG-I and the ATPase deficient mutant (K270A/R) upon stimulation by several RNA substrates under different ATP and ATP analog conditions. The authors have found that ATP binding is sufficient to initiate a signaling response in the presence of an optimal RNA ligand. They provide a biochemical mechanism whereby RIG-I can distinguish between self and non-self RNA, and utilize ATP hydrolysis to rapidly recycle itself, thereby enabling RIG-I to sample the RNA milieu rapidly. By using a variety of substrates and ATP analogs the authors determined that RIG-I utilizes ATP hydrolysis to further distinguish between innate RNA and pathogenic RNA. ATP hydrolysis assists in turning over RIG-I when bound to endogenous RNA by stimulating RNA dissociation. This manuscript contains interesting and important observations in the detailed mechanism of RIG-I activation especially ATP binding and exposure of CARDs.

Overall, the data presentation and writing are extremely clear and logical. The paper presents a detailed biochemical view that nicely complements previous structural work and their cell-based assay provides an excellent platform to test and confirm the biochemical findings. In general, the manuscript is well-written and the data are presented clearly. However, several concerns and points of clarification are listed below.

Essential revisions:

Points of clarification (most do not require further experimentation, and it may be that the authors already have the data that address these questions and comments):

1) Given the important focus of this work on ATP binding vs. ATP hydrolysis, it is surprising that the normal ligand, ATP, was not tested in Figure 4/Table 2. Consequently, to understand and verify the roles of ATP binding and hydrolysis on the release of RNA ligand from RIG-I in Figure 4, authentic ATP needs be examined in addition to ATP analogues.

2) Many previous studies have revealed the importance of ATPase activity for RIG-I signaling. The authors explain how 5'ppp10L acts as an exceptional RNA ligand, acting in an ATP hydrolysis-independent manner. Then, an important question is whether virus-induced RIG-I activation in general, requires hydrolysis or just ATP binding. 5'ppp50L induced a much higher IFN production compared to 5'ppp10L (Figure 1 and Table 1), showing that the internal duplex RNA is important for signaling. The authors need to further address and explain this issue, especially given that in Figure 1, the affinity of WT protein for ATP paradoxically decreases with RNA length.

3) The authors use ATPγS and ADP-AlF_x_ as analogs for ATP. However, it is never established in the manuscript that these are indeed acting as ATP analogs and not, as can often be the case, ADP analogs. In Table 1 (“Equilibrium RNA Binding”), the behavior of ADP-AlF_4_ is identical to that of ADP, and the behavior of ATPγS is less than 2-fold different. The analogs are used to make many of the conclusions regarding the regulatory mechanism, yet they seem to behave like ATP. The authors need to establish that these analog indeed mimic ATP and not ADP. Perhaps this has already been established in the literature. If not, then this could be addressed by competition experiments (e.g., ATPase activity), by the stopped-flow experiments (e.g., as in Figure 2 or 4), or by RNA affinity measurements using the ATPase-dead mutant protein and 5'ppp10L.

4) The experiments establishing ATP binding use MANT-ATP and MANT-ADP. These results with these fluorescent analogs are not adequately documented. Is MANT-ATP a normal substrate for ATP hydrolysis? The *K*_M_ and *k*_cat_ for at least one RNA activator should be measured and reported.

5) Figure 5 shows that RNA ligands with 5'ppp induce a conformational change resulting in CARD exposure, while host cell RNA or internal duplex regions of RNA dissociate RIG-I without exposure of CARD even inducing ATPase activity. The manuscript would be improved if the authors showed the published structure of RIG-I in an additional figure with one or more panels. The structure could be used to point out sites of steric clash between the CARD domain and the CTD domain upon binding to RNA, and help guide the reader through the detailed structural descriptions in the Discussion.

In addition to these essential points, the editors feel that the manuscript could be made more accessible to the general reader by doing the following:

1) Provide an introductory schematic diagram and brief explanation showing the RIG1 cycle. As written, the logic of the paper is difficult for a general reader to follow without first reading a review.

2) In the beginning of the Discussion, provide a brief summary of the expected chemical differences between pathogenic RNA and normal cellular RNA.

Other points to address in preparing the revised manuscript:

A) Regarding Figure 2, none of the stopped flow data are shown. Also, the data in Figure 2 report relative affinities that have been normalized, but the values to which they have been normalized are not provided (relative to Figure 1?). Consequently, the authors need to show an example of the stopped-flow data, the (linear), concentration dependence of MANT-ATP biding, and the results of the fits (values for *k*_on_, *k*_off_, and *K*_d_ for each of the bars in Figure 2) – this could go into supplementary information. Finally, MANT-ADP is mentioned in the Methods, but it does not appear to have been used in the manuscript; if not, then delete or clarify.

B) The number of significant figures in Tables 1 and 2 are unjustifiably too high. In typical biochemical experiments, precision is unlikely to exceed 3 significant figures. For example in Table 1, the first entry for *k*_cat_ is reported as 14.08 +/- 0.37, when it is unlikely to be so precise (and not supported by the S.D.), and a value of 14.1 +/- 0.4 is a more realistic representation of the precision. The comment applies to all of the values in Tables 1 and 2 (a particularly egregious example is the binding constant reported as 41.68 +/- 11.16; this number is no better than 41.7 +/- 11, and perhaps 42 +/- 11 is a more realistic value): unless the authors can demonstrate a higher precision, all of the reported values should be reduced to no more than 3 significant figures. Also, the data in Table 2 for the CARDs lacks SDs, and need to be added.

C) The quality of the RNAs, especially dumbbell RNA, used in this study should be shown.

[Editors' note: further revisions were requested prior to acceptance, as described below.]

Thank you for resubmitting your work entitled “Establishing the role of ATP for the function of the RIG-I innate immune sensor” for further consideration at *eLife*. Your revised article has been favorably evaluated by John Kuriyan (Senior Editor) and a Reviewing Editor. The manuscript has been improved but there are some remaining issues that need to be addressed before acceptance, as outlined below:

It appears that you have addressed all of the reviewers and editors comments except one, which is comment #1. This was a most important point that all of the reviewers felt needed to be addressed experimentally. All were aware of the complexity that ATP hydrolysis introduces, and that it was certainly not possible for equilibrium experiments with WT protein. But all nonetheless felt it was critically important to include such experiments with authentic ATP where feasible. Old Figure 4 (now 5) shows equilibrium binding data for two ATP-hydrolysis defective mutants in panels A and C (K270A and K270R); equilibrium data with ATP can unquestionably be obtained for these two mutant proteins, and the interpretation of those binding data will not be encumbered with the complexities of steady-state ATP hydrolysis. In addition, the kinetic experiments in panels B and D can be performed with both WT and mutant protein. The detailed kinetic analysis of the rates and amplitudes for WT protein can be deferred to another paper, but the observed behavior is important to report, relative to the analogs. Again, however, the interpretation of the kinetic behavior of the ATP-hydrolysis defective mutants should be straightforward, given the model and the nil turnover relative to the dissociation time; consequently these results also need to be provided.

---

## [Author Response]

Essential revisions:

Points of clarification (most do not require further experimentation, and it may be that the authors already have the data that address these questions and comments):

*1) Given the important focus of this work on ATP binding vs. ATP hydrolysis, it is surprising that the normal ligand, ATP, was not tested in*
Figure 4*/*Table 2*. Consequently, to understand and verify the roles of ATP binding and hydrolysis on the release of RNA ligand from RIG-I in*
Figure 4*, authentic ATP needs be examined in addition to ATP analogues.*

We agree that it would be ideal to monitor RNA dissociation rate and equilibrium constants in the presence of ATP, however the lability of ATP makes such measurements impossible. A significant amount of ATP would be hydrolyzed over the time scale of the experiment (up to 10 minutes in the kinetic assays, and longer in the case of equilibrium measurements), meaning that any measured dissociation would reflect RIG-I in ATP, ADP+Pi, and ADP bound states. As we cannot accurately determine the proportions of these states, we cannot quantitate a rate constant for ATP-bound RIG-I, and instead would be reporting on rate and equilibrium constants for a heterogeneous mixture of nucleotide-bound RIG-I molecules.

*2) Many previous studies have revealed the importance of ATPase activity for RIG-I signaling. The authors explain how 5'ppp10L acts as an exceptional RNA ligand, acting in an ATP hydrolysis-independent manner. Then, an important question is whether virus-induced RIG-I activation in general, requires hydrolysis or just ATP binding. 5'ppp50L induced a much higher IFN production compared to 5'ppp10L (*Figure 1
*and*
Table 1*), showing that the internal duplex RNA is important for signaling. The authors need to further address and explain this issue, especially given that in*
Figure 1*, the affinity of WT protein for ATP paradoxically decreases with RNA length.*

Thank you for your comment. It is important to note that, irrespective of any findings involving 5’ppp50L, our manuscript provides sufficient mutational evidence that ATP binding rather than hydrolysis is required for signaling in the biological context. Using the K270A mutant, we show that hydrolysis is dispensable for RIG-I activation, meaning that the physical transition from an autorepressed state to an activated, signaling competent state does not depend on the ability of the protein to hydrolyze ATP.

Regarding the signaling activity of 5’ppp50L, we would respectfully disagree that 5’ppp50L supports “much higher IFN production” than 5’ppp10L. As shown in Table 1, it is only two-fold higher (46 vs. 23 units), and with another long ligand (5’ppp30L) IFN production is actually lower (16 units) than 5’ppp10L. Thus we are not consistently seeing better signaling from longer ligands, and while the effect of 5’ppp50L is significant and reproducible, it would not be considered a major effect. However, to address the basis for the higher observed signaling on 5’ppp50L, we would reference previous reports from the Hur and Garcia-Sastre labs, which have shown that the RIG-I protein is capable of forming multimers on longer RNA ligands (33; 35). Given the small effect we are reporting here, we attribute the higher levels of IFN production induced by a sufficiently long RNA (in this case 5’ppp50L) to local concentration effects that are caused by multimerization of RIG-I on a single RNA ligand (new data shown as Figure 3—figure supplement 1).

Importantly, it has been claimed that 4 copies of the RIG-I CARDs protein must form specific intermolecular tetramers in order to nucleate immune signaling through the MAVS protein (34), however we only observe a 3:1 complex between RIG-I and 5’ppp50L, suggesting that enhanced signaling induced by this ligand cannot be attributed to a tetrameric complex formation on a single RNA. We also show that an individual RIG-I molecule bound at an internal site is incapable of inducing an immune response (Figure 4). Thus, while we appreciate your point, we respectfully disagree that internal duplex RNA is important for signaling. Instead, we would argue that internal sites may facilitate *enhanced* signaling by RIG-I under conditions of high protein concentration, and it is unlikely that these sites are required or important for signaling in a biological context.

*3) The authors use ATPγS and ADP-AlF*_*x*_
*as analogs for ATP. However, it is never established in the manuscript that these are indeed acting as ATP analogs and not, as can often be the case, ADP analogs. In*
Table 1
*(“Equilibrium RNA Binding”), the behavior of ADP-AlF*_*4*_
*is identical to that of ADP, and the behavior of ATPγS is less than 2-fold different. The analogs are used to make many of the conclusions regarding the regulatory mechanism, yet they seem to behave like ATP. The authors need to establish that these analog indeed mimic ATP and not ADP. Perhaps this has already been established in the literature. If not, then this could be addressed by competition experiments (e.g., ATPase activity), by the stopped-flow experiments (e.g., as in*
Figure 2
*or 4), or by RNA affinity measurements using the ATPase-dead mutant protein and 5'ppp10L.*

Thank you for this point and for the suggested experiments. There is existing data indicating that the ATP analogs differ in several key ways from ADP, but the competition experiment remains a good idea (see below). For example, the data in Figure 4 (now Figure 5) demonstrate that ATPγS behaves reproducibly differently than ADP and ADP-AlF_4_. In addition, we would draw the reviewer’s attention to the published structural data from Stephen Cusack’s group in which RIG-I was crystallized bound to RNA and ADP-AlF_4_ in a conformation distinct from that observed in structures where the protein was crystallized with RNA and ADP, as published by both of our labs ([23]; Luo et al., 2012).

To compare the binding of the ATP analogs with ADP, we performed competition experiments in which the ATP hydrolysis activity of RIG-I was evaluated in the presence of ATP and each of the nucleotide analogs that were used in this study (new data shown as Figure 5—figure supplement 1). As expected, ADP, ATPγS, ADP-AlF_4_, and AMPPNP all compete effectively with ATP for binding the RIG-I active site, resulting in diminished rate constants of ATP hydrolysis. Additionally, we observe that ADP most strongly inhibits RIG-I ATPase activity, while ATPγS and AMPPNP inhibit activity more weakly and to similar levels, and ADP-AlF_4_ exhibits the weakest degree of inhibition. This is once again consistent with a scenario in which all analogs bind at the RIG-I active site, and behave distinctly from ADP. The behavior of ADP in this system is inherently interesting, and experiments on that topic are the subject of another paper that we are presently working on.

*4) The experiments establishing ATP binding use MANT-ATP and MANT-ADP. These results with these fluorescent analogs are not adequately documented. Is MANT-ATP a normal substrate for ATP hydrolysis? The* K_*M*_
*and* k_*cat*_
*for at least one RNA activator should be measured and reported.*

To demonstrate that MANT-ATP acts as a functional substrate for RIG-I, we measured the ability of RIG-I bound to 5’ppp10L to stimulate hydrolysis of MANT-ATP as a function of MANT-ATP concentration (new data shown as Figure 2—figure supplement 1). We observed MANT-ATP hydrolysis with a *k*_cat_ = 3.9 ± 0.2 s^-1^ and *K*_m,MANT-ATP_ =115 ± 25 μM. Importantly, RIG-I exhibits a similar Michaelis constant for MANT-ATP and ATP, indicating that the protein binds both substrates with similar affinity, making MANT-ATP a good mimetic of ATP binding. However, the lower *k*_cat_ determined for MANT-ATP suggests that addition of the fluorescent moiety impedes hydrolysis. This phenomenon works in our favor, as it means that vanishingly little MANT-ATP will be hydrolyzed over the time-scale of our experiments, ensuring that we are measuring binding of an ATP and not an ADP mimetic.

*5)*
Figure 5
*shows that RNA ligands with 5'ppp induce a conformational change resulting in CARD exposure, while host cell RNA or internal duplex regions of RNA dissociate RIG-I without exposure of CARD even inducing ATPase activity. The manuscript would be improved if the authors showed the published structure of RIG-I in an additional figure with one or more panels. The structure could be used to point out sites of steric clash between the CARD domain and the CTD domain upon binding to RNA, and help guide the reader through the detailed structural descriptions in the Discussion.*

We have added a structure-based diagram as Figure 6 to help clarify the process of RIG-I compaction and CARDs/CTD collision that we propose promotes signal activation or dissociation.

In addition to these essential points, the editors feel that the manuscript could be made more accessible to the general reader by doing the following:

1) Provide an introductory schematic diagram and brief explanation showing the RIG1 cycle. As written, the logic of the paper is difficult for a general reader to follow without first reading a review.

We have provided a simple RIG-I activation scheme and included it in our manuscript as Figure 1.

2) In the beginning of the Discussion, provide a brief summary of the expected chemical differences between pathogenic RNA and normal cellular RNA.

We have addressed and provided references for the expected differences between endogenous RNAs and those deposited or transcribed by viruses that activate RIG-I in both the Introduction and Discussion sections of the manuscript.

Other points to address in preparing the revised manuscript:

*A) Regarding*
Figure 2*, none of the stopped flow data are shown. Also, the data in*
Figure 2
*report relative affinities that have been normalized, but the values to which they have been normalized are not provided (relative to*
Figure 1*?). Consequently, the authors need to show an example of the stopped-flow data, the (linear), concentration dependence of MANT-ATP biding, and the results of the fits (values for* k_*on*_*,* k_*off*_*, and* K_*d*_
*for each of the bars in*
Figure 2*) – this could go into supplementary information. Finally, MANT-ADP is mentioned in the Methods, but it does not appear to have been used in the manuscript; if not, then delete or clarify.*

We have clarified the method of data normalization for Figure 2 (now Figure 3) in the figure legend. We have also produced a supplemental figure showing representative traces of MANT-ATP association for each construct where binding was observed (Figure 2—figure supplement 1). We show linear fits of observed association rates from which *k*_on_, *k*_off_, and *K*_d_ were derived (Figure 2—figure supplement 1).

*B) The number of significant figures in*
Tables 1 and 2
*are unjustifiably too high. In typical biochemical experiments, precision is unlikely to exceed 3 significant figures. For example in*
Table 1*, the first entry for kcat is reported as 14.08 +/- 0.37, when it is unlikely to be so precise (and not supported by the S.D.), and a value of 14.1 +/- 0.4 is a more realistic representation of the precision. The comment applies to all of the values in*
Tables 1 and 2
*(a particularly egregious example is the binding constant reported as 41.68 +/- 11.16; this number is no better than 41.7 +/- 11, and perhaps 42 +/- 11 is a more realistic value): unless the authors can demonstrate a higher precision, all of the reported values should be reduced to no more than 3 significant figures. Also, the data in*
Table 2
*for the ΔCARDs lacks SDs, and need to be added.*

Thank you for noticing this issue and bringing it to our attention. We have modified our reported figures to reflect a biochemically appropriate number of significant digits.

C) The quality of the RNAs, especially dumbbell RNA, used in this study should be shown.

We have run a non-denaturing polyacrylamide gel to visualize purity of major RNA ligands and included it as Figure 3—figure supplement 2. For the dumbbell RNA, we have also included the labeling efficiency control experiment (Figure 3—figure supplement 2) originally referenced in the Materials and methods section in the same figure.

[Editors' note: further revisions were requested prior to acceptance, as described below.]

*It appears that you have addressed all of the reviewers and editors comments except one, which is comment #1. This was a most important point that all of the reviewers felt needed to be addressed experimentally. All were aware of the complexity that ATP hydrolysis introduces, and that it was certainly not possible for equilibrium experiments with WT protein. But all nonetheless felt it was critically important to include such experiments with authentic ATP where feasible. Old*
Figure 4
*(now 5) shows equilibrium binding data for two ATP-hydrolysis defective mutants in panels A and C (K270A and K270R); equilibrium data with ATP can unquestionably be obtained for these two mutant proteins, and the interpretation of those binding data will not be encumbered with the complexities of steady-state ATP hydrolysis. In addition, the kinetic experiments in panels B and D can be performed with both WT and mutant protein. The detailed kinetic analysis of the rates and amplitudes for WT protein can be deferred to another paper, but the observed behavior is important to report, relative to the analogs. Again, however, the interpretation of the kinetic behavior of the ATP-hydrolysis defective mutants should be straightforward, given the model and the nil turnover relative to the dissociation time; consequently these results also need to be provided.*

To address these concerns, we performed additional equilibrium RNA binding assays and we monitored RIG-I dissociation from RNA in the presence and absence of ATP, comparing the results with data obtained in the presence of ATPγS and ADP. In equilibrium binding experiments using ATPase-dead mutants K270A and K270R, the presence of ATP weakened the *K*_d_ for RNA by between 2 and 2.5 fold, resulting in values similar to those we previously reported for ATPγS (Figure 8).

Author response image 1.Relative equilibrium binding affinities of Walker A mutant RIG-I for 5’OH10L in the absence of nucleotide (RNA), or with ATP or ATPγS. All affinities are normalized to mean RNA-only affinity. Error bars represent standard deviation (n=3).**DOI:**
http://dx.doi.org/10.7554/eLife.09391.018

In experiments designed to measure rate constants for RNA dissociation from the K270A and K270R mutants, the effect of ATP was also similar to that observed for ATPγS (Figure 9). In the case of WT RIG-I, the inclusion of ATP results in a small enhancement of RNA *k*_off_ relative to the values obtained for ATPγS and ADP. This may be attributed to the mixture of RIG-I populations (i.e. RIG-I bound to substrate, transition state and product) undergoing active hydrolysis over the time period of data collection (5-15 min.), perhaps resulting in significant population of the transition state conformation. Consistent with this, a quantitatively similar enhancement in *k*_off_ is observed in the presence of the transition state analog, ADP-AlF_4_ (see Table 3 below). Indeed, dissociation rate constants for WT RIG-I in the presence of ADP-AlF_4_ and ATP are similar.

Author response image 2.RNA dissociation rate constant determination of Walker A mutant RIG-I from 5’ppp10L in the absence (Control, black) or presence of ATP (green) or ATPγS (red). Traces represent an average over 4-6 trials.**DOI:**
http://dx.doi.org/10.7554/eLife.09391.019



Author response table 1.RNA dissociation rate constants and equilibrium RNA binding affinities for WT and mutant RIG-I constructs.**DOI:**
http://dx.doi.org/10.7554/eLife.09391.020*RNA Binding Interference Data**Protein**Nucleotide**k*_1_ (x 10^-3^ sec^-1^)Amp.*k*_2_ (x 10^-3^ sec^-1^)Amp.*K*_d, 5’OH10L_WT RIG-INone28 ± 1325%4.0 ± 0.275%17 ± 1.0ATP244 ± 397%42 ± 83%--ADP-AlF_4_206 ± 372%32 ± 0.628%34 ± 4.9ATPγS187 ± 345%26 ± 0.455%50 ± 5.0ADP187 ± 340%25 ± 0.360%32 ± 2.0K270ANone108 ± 180%5.3 ± 0.120%42 ± 11ATP327 ± 279%25 ± 0.321%83 ±7ATPγS390 ± 375%35 ± 0.425%98 ± 8.4K270RNone92.9 ± 153%16 ± 0.147%30 ± 4ATP227 ± 270%14 ± 0.130%82 ± 2ATPγS224 ± 181%16 ± 0.219%66 ± 6.1ΔCARDsNone31 ± 0.534%2.1 ± 0.0266%19 ± 1.0ATP27 ± 0.230%1.2 ± 0.0170%--ATPγS24 ± 0.327%2.2 ± 0.0173%28 ± 0.7ADP35 ± 0.523%1.7 ± 0.0177%24 ± 2.3

When ATP was included in *k*_off_^RNA^ experiments using the ΔCARDs construct, we did not observe an enhancement of the RNA dissociation rate, consistent with earlier observations made with ATPγS and ADP (see table below). We feel that these results demonstrate that the binding interaction between RIG-I and ATPγS is functionally highly similar to the interaction between RIG-I and ATP, and that they argue in favor of our original model for the role of ATP binding in RIG-I ligand selection.

These findings are addressed within the manuscript in the subsection “Nucleotide binding stimulates RNA dissociation by RIG-I”, wherein we reference the above experiments and interpret our results in the context of data obtained with ADP, ADP-AlF_4_ and ATPγS. We have included equilibrium binding constants and dissociation rate constants for experiments performed with the K270A and K270R mutant constructs in Tables 1 and 2 respectively. If you believe it is necessary, we can also produce an additional figure supplement showing the quantitated equilibrium binding data and dissociation traces for the mutant proteins (as in Figure 8 and Figure 9); however we felt that the inclusion of the measured biophysical constants would be sufficient.